# Generalized cue reactivity in rat dopamine neurons after opioids

Collin M. Lehmann[1], Nora E. Miller[1], Varun S. Nair[1], Kauê M. Costa ®[2], Geoffrey Schoenbaum ®[3] & Khaled Moussawi ®[1,4] ✉

Cue reactivity is the maladaptive neurobiological and behavioral response upon exposure to drug cues and is a major driver of relapse. A widely accepted assumption is that drugs of abuse result in disparate dopamine responses to cues that predict drug vs. natural rewards. The leading hypothesis is that drug-induced dopamine release represents a persistently positive reward prediction error that causes runaway enhancement of dopamine responses to drug cues, leading to their pathological overvaluation. However, this hypothesis has not been directly tested. Here, we develop Pavlovian and operant procedures in male rats to measure firing responses within the same dopamine neurons to drug versus natural reward cues, which we find to be similarly enhanced compared to cues predicting natural rewards in drug-naive controls. This enhancement is associated with increased behavioral reactivity to the drug cue, suggesting that dopamine neuronal activity may still be relevant to cue reactivity, albeit not as previously hypothesized. These results challenge the prevailing hypothesis of cue reactivity, warranting revised models of dopaminergic function in opioid addiction, and provide insights into the neurobiology of cue reactivity with potential implications for relapse prevention.

Enduring relapse vulnerability remains a major challenge for the treatment of substance use disorders[1] and attempts to curb relapse rates have not yielded significant improvements in the last fifty years[2]. Exposure to drug-associated cues triggers craving and increases relapse risk, even during prolonged abstinence[3]. This reflects the enhanced and enduring motivational effect of these cues[4]. Such neurobiological and behavioral response is referred to as cue reactivity. Several models of addiction emphasize the role of cue reactivity and propose that addiction is a dopamine-dependent disorder of associative learning whereby repeated exposure to addictive drugs results in overvaluation and overpowering salience of drug cues through abnormally strong and long-lasting cue-drug associations[5–9].

Dopamine neurons provide a teaching signal that resembles reward prediction errors and modulates cue-reward associations[10–13]. A leading hypothesis of cue reactivity is that addictive drugs, all of which increase dopamine signaling due to their direct pharmacological effects even when the cue-drug association is fully learned[14], result in enhanced maladaptive learning by causing a persistently positive reward prediction error every time the drug is taken[5,6,15]. Over time, this leads to a runaway enhancement of dopamine firing response to drug cues that approaches a maximum, resulting in overvaluation and triggering intense craving and relapse. Critically, this hypothesis posits a selective enhancement of dopamine response to drug cues, to the exclusion of other, non-drug cues. While this account is parsimonious, it has been subject to criticism; data show that the blocking effect – the effect of previous learning about a cue to prevent new learning about another cue – continues to occur with addictive drugs[16,17], unlike what is predicted by the noncompensable reward prediction error hypothesis. Also, smaller-than-expected drug rewards result in reduced lever pressing suggesting that negative prediction error could occur for drugs[18]. Although the reward prediction error-based theory's

[1]Department of Psychiatry, University of Pittsburgh, Pittsburgh 15219, USA. [2]Department of Psychology, University of Alabama at Birmingham, Birmingham 35233, USA. [3]National Institute on Drug Abuse, National Institutes of Health, Baltimore 21224, USA. [4]Department of Neurology, University of California San Francisco, San Francisco 94158, USA. ✉e-mail: khaled.moussawi@ucsf.edu

parsimony derives from its adherence to the temporal difference model of dopamine, this model itself has been criticized, as new evidence has shown that dopamine signals reflect computations that incorporate a richer set of information than the traditional temporal difference account allows. Accordingly, several modified temporal difference and non-temporal difference alternative models have been proposed[18–22] some of which complicate or entirely contradict this reward prediction error model of addiction. Further, the core prediction that dopamine reward prediction error responses to drug-related cues should be selectively increased has not been tested. It is this prediction which links the molecular dynamics of dopamine to the behavioral disorder of addiction, and which is therefore most critical to experimentally challenge. We conducted, across different institutions, two independent experiments using Pavlovian and operant procedures to directly test this prediction. We used single-unit recordings in rat ventral tegmental area (VTA) to compare dopaminergic firing responses (hereafter dopamine response) within the same neurons to cues associated with opioids (remifentanil) vs. natural rewards.

Here, we show that within-neuron responses are not different between cues predicting opioids vs. natural rewards in animals with a history of opioid exposure. Our data do not support the assumption that addiction ensues from differential dopamine signaling to drug vs. non-drug cues, challenging the noncompensable reward prediction error hypothesis of cue reactivity and suggesting a different understanding of how dopamine neurons respond to cues and rewards after repeated opioid exposure.

## Results
### Pavlovian procedure (Experiment 1)
We first validated a rat model that allows simultaneous single-unit recordings and intravenous (IV) delivery of the ultrashort-acting

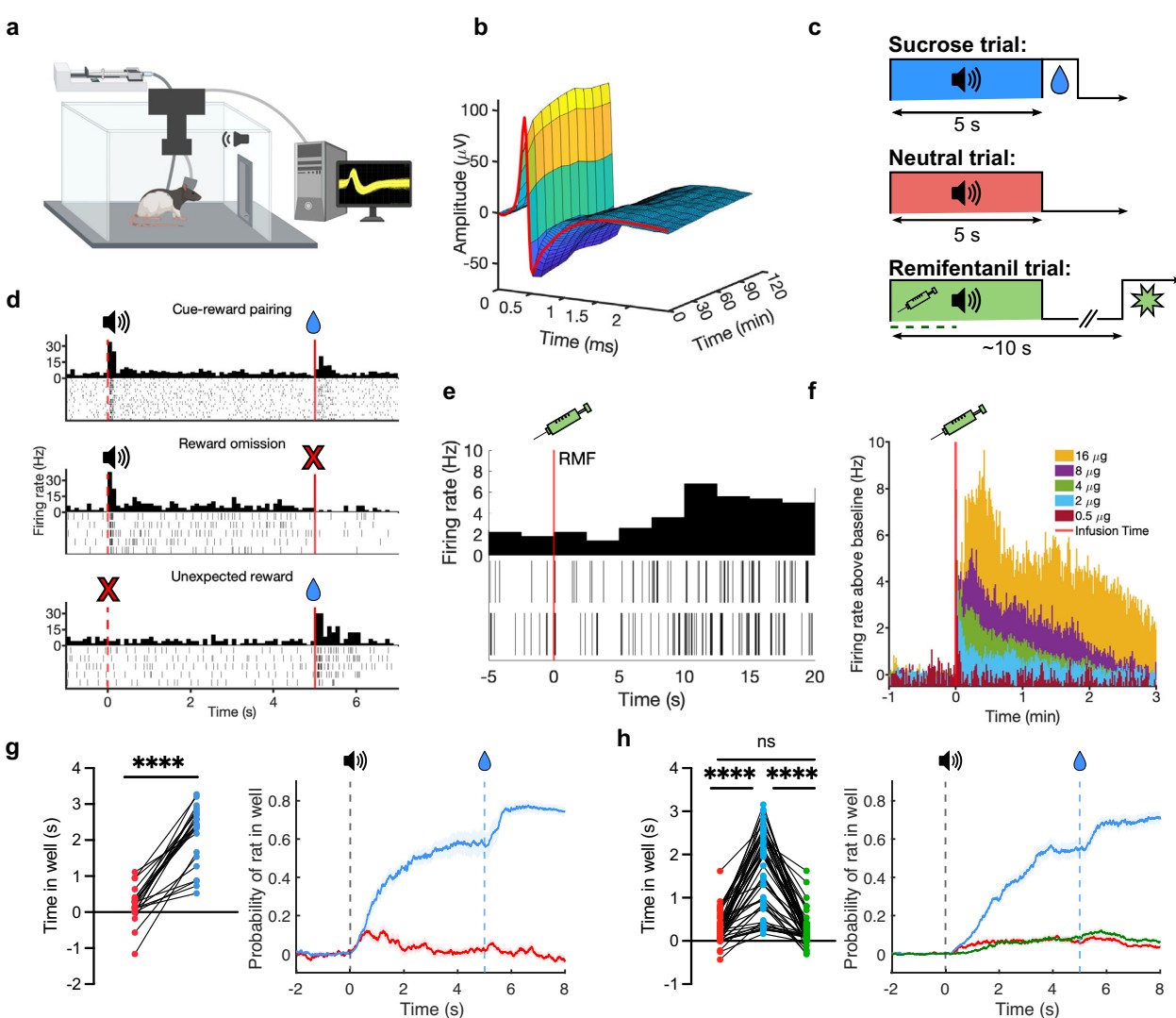

**Fig. 1 | Pavlovian experimental procedure. a** Schematic of operant chamber for training/recording in Experiment 1[76]. **b** Example waveform showing stability over a two-hour recording session. Color corresponds to action potential amplitude. **c** Pavlovian conditioning of three 5-s tones predicting sucrose (blue), remifentanil (green), or nothing (neutral, red). **d** Example responses of putative dopamine responses to cued sucrose reward (top), reward omission (middle), and uncued reward (bottom). **e** Rasters and post-stimulus time histogram (PSTH) showing example dopamine neuron firing after remifentanil infusion (2 infusions of 4 μg/kg remifentanil) with onset of remifentanil effect at -10 s post-infusion. **f** PSTH for neuronal data of onset and duration of effect of remifentanil at doses of 0.5 to 16 μg/kg/infusion prior to training (activation terminates by -2.5 min for 4 μg/kg). **g** Total time (left) and probability (right) of rat in sucrose reward well during sucrose and neutral cue presentations in opioid-naive rats ($N = 4$ rats, $n = 22$ units, two-tailed Wilcoxon test, $z = -2.105$, $p < 0.0001$). Lines and shades represent means ± SEM. **h** Same as h but in opioid-exposed animals after sucrose, remifentanil, and neutral cues ($N = 7$ rats, $n = 50$ units, Friedman test, $F = 75.36$, $p < 0.0001$ with Dunn's multiple comparison test: sucrose vs. remifentanil $p > 0.9999$, sucrose vs. neutral $p < 0.0001$, remifentanil vs. neutral, $p < 0.0001$). Source data are provided as a Source Data file.

opioid remifentanil (RMF) (Fig. 1). Remifentanil is a selective μ opioid agonist with similar reinforcing properties to other opioids[23] and a short half-life (0.3–1.1 min)[24]. Rats ($n = 11$) were implanted with drivable microelectrode bundles targeting the VTA, and jugular catheters for IV fluid delivery. Rats were trained in a behavior box with a customized commutator to prevent tangling of the catheter line and headstage cable allowing for stable extended recording sessions (Fig. 1a, b).

Because the tested hypothesis predicts that the drug cue response would be greater than the natural reward cue response regardless of value, we used sucrose as the natural reward given its high rewarding properties compared to water[25], to minimize the possibility of observing the predicted pattern simply because of a lower relative value of the natural reward. Rats were water-restricted and trained with three trial types in each session, each associating a distinct 5-s auditory cue with a unique outcome (Fig. 1c). The dopamine responses to oral sucrose and IV remifentanil rewards cannot be directly compared, as responses to sucrose reward are precise and discrete (time-locked within the first 1 s following delivery) (Fig. 1d), whereas the firing response to remifentanil is diffuse, delayed, and variable, peaking and diffusing over tens of seconds (Fig. 1e, f). However, the model we test predicts that remifentanil inevitably evokes reward prediction errors that uniquely escalate to maximal drug-cue response, compared to natural-reward cue, regardless of the specific quantity of the remifentanil used, so such matching is unnecessary. Thus, we simply chose sucrose volumes and remifentanil doses that are effective reinforcers[23]. Cue 1 (henceforth 'sucrose cue') was immediately followed by the delivery of a bolus of sucrose at the designated well (40 μL). Cue 2 ('remifentanil cue') was presented simultaneously with activation of the infusion pump for IV remifentanil delivery (4 μg/kg/infusion). Cue 3 ('neutral cue') resulted in no consequence. A subset of rats ($n = 4$) did not receive any remifentanil (opioid-naive comparison group) and were only presented with sucrose and neutral cues. Following remifentanil infusion, the drug was allowed to clear for approximately 110–320 s (mean = 173 s) based on the observed timecourse of the remifentanil effect on neuronal firing (Fig. 1f) before initiation of the next trial. Successful discrimination of cue identities was demonstrated by different cue-induced responding, showing significantly greater probability of entering the sucrose well during sucrose vs. remifentanil or neutral cues (Fig. 1g, h).

Putative dopamine neurons were identified using hierarchical clustering that has been shown to accurately discriminate genetically identified dopamine neurons[13,26–28] based on unit activity during sucrose trials (Fig. 2a), and refined by $k$-means clustering (Supplementary Fig. 1). Identification based on waveform properties and D2 inhibition of select units was more conservative, but otherwise largely agreed with these results. Hierarchical clustering identifies canonical dopamine neurons based on their firing response to cues and rewards. More traditional dopamine identification methods based on waveform criteria (e.g., initial waveform positivity and waveform duration) and inhibition by a D2 agonist are effective at screening out non-dopaminergic neurons but have been shown to be overly conservative or non-specific[26,29,30]. For example, many dopamine neurons do not express D2 receptors and some genetically identified non-dopamine neurons are inhibited by D2 agonists[31]. Dopamine neurons identified with the activity-based clustering showed longer duration and higher probability of D2 inhibition than non-dopamine neurons. Similarly, $k$-means clustering based on waveform properties (amplitude-ratio and half-duration)[19] yielded concordant but more conservative identification of dopamine neurons (Supplementary Fig. 2). However, many likely dopamine neurons were excluded with the waveform-based clustering. One neuron was captured in two channels (nearly one-to-one correspondence of spikes between channels) and was classified as likely dopamine neuron based on waveform criteria

in one channel, but non-dopamine in the other (Supplementary Fig. 2). Such differences in waveform properties are likely due to the placement of the recording wires relative to the neuron. Hierarchical activity-based clustering avoids this problem.

## Within-neuron responses Pavlovian cues

Contrary to the noncompensable reward prediction error hypothesis of cue reactivity, we found no significant difference in the responses of dopamine neurons to cues predicting sucrose (40 μL) vs. remifentanil (4 μg/kg) cues ($n = 21$) (Fig. 2c, d), despite the rats being able to discriminate between these cues (Fig. 1h). Average firing did not differ between sucrose and remifentanil cues over the 500 ms following cue onset (Fig. 2b–d). We found that sucrose, remifentanil, and neutral cues elicited significant responses in the first 30-180 ms after cue onset (detection component), but only rewarded cues continued to be excited over the next 320 ms (evaluation component), consistent with a two-component model[32] (Fig. 2b–d). However, we found no differences between remifentanil and sucrose cues in either phase (Fig. 2e–g). In addition, if remifentanil-associated cues tended to elicit a greater dopamine response than sucrose cues regardless of dose, then we would expect a greater number of neurons to exhibit a higher firing rate in response to remifentanil than sucrose cues. However, a within-unit comparison of both responses found no significant difference in the numbers of remifentanil and sucrose-preferring neurons in both the detection (Fig. 2f) and evaluation (Fig. 2g) phases. Both sucrose and remifentanil reward were associated with increased dopamine response, but over distinct timescales, with remifentanil response distributed with less moment-to-moment activation over substantially longer duration (Supplementary Fig. 3).

## Sensitized dopamine responses

We compared units obtained from opioid-exposed rats ($n = 77$, $N = 4$) to those from opioid-naive rats ($n = 22$, $N = 4$) (Fig. 3) and observed that the average baseline firing rate measured before any opioid administration was greater for opioid-exposed than opioid-naive units (Fig. 3c). Both opioid-exposed and opioid-naive units showed greater responses to cues predicting 40 μL of sucrose vs. neutral cues, and a significant, positive response to reward delivery simultaneous with sucrose cue offset, but not to neutral cue offset (Fig. 3e, f). However, opioid-exposed units showed significantly greater responses to sucrose and neutral cues (Fig. 3g, h) and sucrose delivery (Fig. 3i) than opioid-naive units. The amount of prior training measured in previous sucrose trials was similar between opioid-naive and opioid-exposed groups (Fig. 3d), so it is unlikely that any differences in these populations are attributable to differences in duration or experience in training.

## Intact natural-reward omission response

In sessions including omission trials (sucrose cue presented without subsequent reward in 10% of trials), both opioid-exposed and opioid-naive units showed a small negative change in firing at the time of expected sucrose delivery (around 500 ms following cue offset) (Fig. 3j). However, opioid-exposed units exhibited a large positive peak (period 1) before the negative component of the omission response (period 2) (Fig. 3j). Period 1 response likely corresponds to cue offset that also indicates reward availability on rewarded trials[33,34], and was significantly greater in opioid-exposed than opioid-naive units, again reflecting the enhanced salience of the sucrose cue and initial expectation of reward delivery in this group. However, period 2 response was similar between groups (Fig. 3k) suggesting intact negative reward prediction error to natural rewards in opioid-exposed subjects. No negative reward prediction error was observed for remifentanil omission likely due to the diffuse nature of the remifentanil-induced firing response.

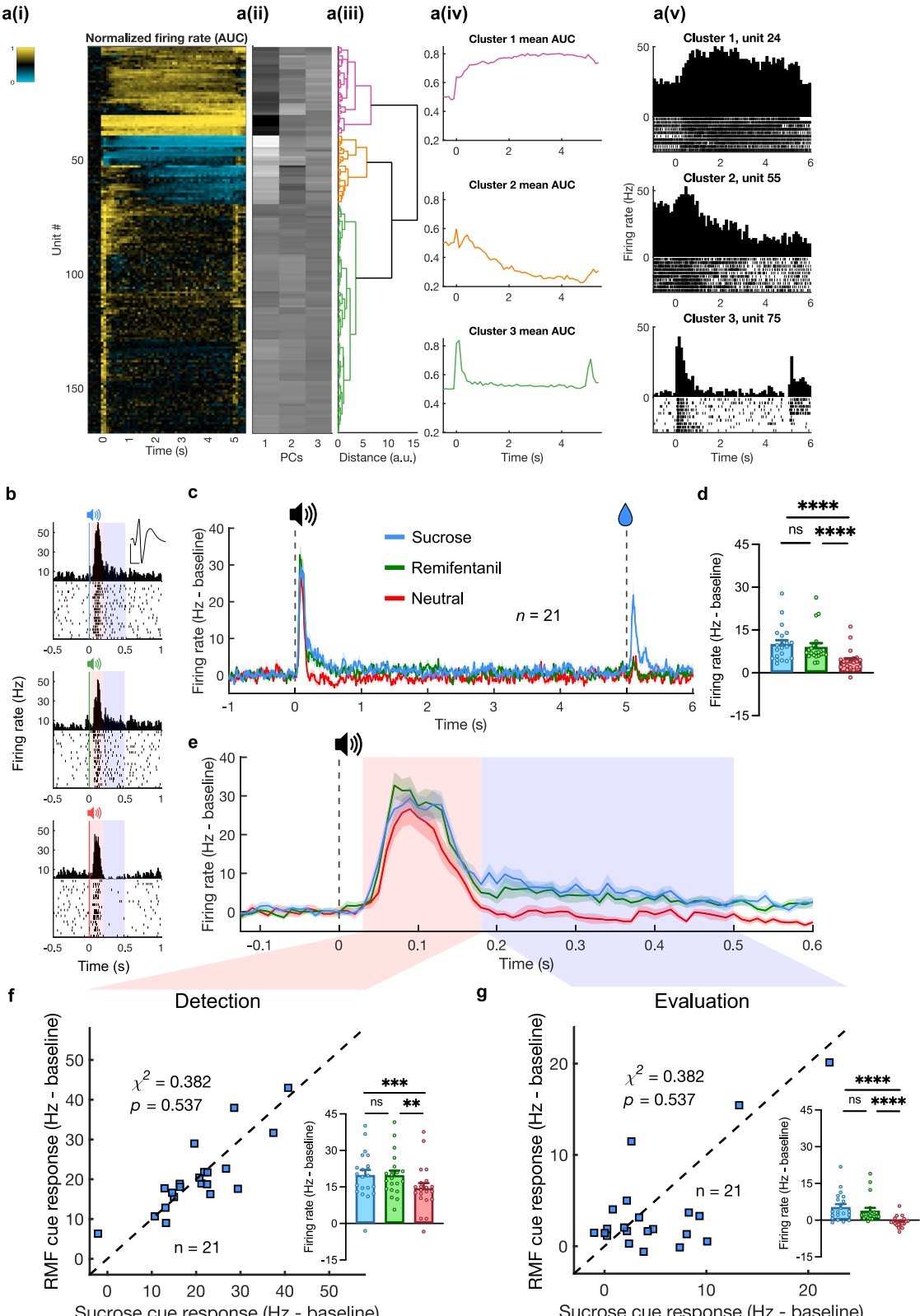

## Progressive ratio for remifentanil vs. sucrose

Considering the above results, we sought to verify that our chosen dose of remifentanil (4 µg/kg) was associated with a greater behavioral motivation than sucrose (40 µL) in line with its addictive potential. Therefore, at the end of Experiment 1, rats were trained on a progressive ratio schedule of reinforcement for sucrose (40 µL) and

remifentanil (4 µg/kg). We found a significantly higher breakpoint and total responses recorded for remifentanil reward vs. sucrose (Supplementary Fig. 4), confirming that the used dose of remifentanil was of higher motivational value than sucrose.

Results of Experiment 1 show that in opioid-exposed subjects, dopamine neurons exhibit higher baseline activity and generalized

**Fig. 2 | Similar within-neuron dopamine firing responses to sucrose and remifentanil predictive cues. a** Dopamine neuron identification (*N* = 11 rats). (i) Heatmap showing functional activation of each unit (rows) aligned to sucrose cue onset (0 s). Activity normalized using area under receiver-operator curve (auROC) method, compared against baseline. Scales from 0 to 1. (ii) First three components extracted via principal component analysis. (iii) Dendrogram showing results from hierarchical clustering. (iv) Mean auROC for units classified into each cluster. (v) PSTH and raster plots from example units for each cluster. **b** Example responses from a single unit to sucrose (blue, top), remifentanil (green, middle), and neutral (red, bottom) cues. Red and blue shaded regions indicate distinct phases of cue response reflecting detection (30–180 ms) and valuation (180–500 ms). Inset waveform represents mean of sorted unit. Scale bars: 50 μV and 500 ms. **c** PSTH traces showing average dopamine firing during sucrose (blue, 40 μL), remifentanil (green, 4 μg/kg), and neutral trials (red). All traces show mean ± SEM. **d** Mean baseline-subtracted firing rates during cue response to sucrose, RMF, and neutral

cues (Friedman test, *F* = 31.71, *n* = 21, *p* < 0.0001, with Dunn's multiple comparison test: sucrose vs. remifentanil *p* > 0.9999, sucrose vs. neutral *p* < 0.0001, remifentanil vs. neutral, *p* < 0.0001). All bars show mean + SEM. **e** Close-up of cue period from Fig. 2c. **f** Scatter plot of individual units' responses to sucrose (abscissa) and RMF (remifentanil; ordinate) cues during the detection phase. Inset: cue responses during the detection phase (Friedman test, *F* = 16.29, *n* = 21, *p* = 0.0003, with Dunn's multiple comparison test: sucrose vs. remifentanil *p* > 0.9999, sucrose vs. neutral *p* = 0.0006, remifentanil vs. neutral, *p* = 0.0036). **g** Individual units' responses during the evaluation phase presented as in Fig. 2f. Note that identical Chi-squared statistics to 2 f are due to equal numbers of units exhibiting firing to remifentanil > sucrose in both intervals. Inset: cue responses during the evaluation phase Friedman test, *F* = 28.67, *n* = 21, *p* < 0.0001, with Dunn's multiple comparison test: sucrose vs. remifentanil *p* > 0.9999, sucrose vs. neutral *p* < 0.0001, remifentanil vs. neutral, *p* < 0.0001). Source data are provided as a Source Data file.

enhanced response to both drug and non-drug cues, and to natural rewards. These findings directly challenge the non-compensable reward prediction error model of cue reactivity.

## Operant procedure (Experiment 2)

Experiment 1 is limited by the absence of behavioral measures of drug-cue reactivity, difficulty in directly comparing reward values and corresponding neural responses, and the potential latent pharmacological effect of remifentanil during subsequent trials. Thus, in a separate group of rats, we conducted Experiment 2 which is based on an operant task whereby distinct discriminative cues predicted the availability of a non-drug or drug reward (trial types A and B, respectively). Operant chambers were equipped with an overhead light, speaker, retractable lever, and two feeders for water delivery outfitted with lights and photobeam entry detectors. Each trial began with the illumination of an overhead light followed by extension of a lever. Following lever press, the trial type was signaled by one of three distinct auditory cues; A, B, or C. Cue C indicated no reward and terminated the trial. Cue A was associated with a water reward and Cue B was associated with an identical water reward at the other port and a simultaneous remifentanil infusion. After cue offset, the rat had to enter the correct feeder associated with the cue played to receive a reward, either water alone or water + remifentanil. Incongruent port entry following cue offset resulted in termination of the trial (Fig. 4a, b). As in Experiment 1, a subset of rats remained opioid-naive, receiving IV saline instead of remifentanil. This design controls for the sensory and temporal properties of the non-drug reward and allows direct comparison of the reward values (value$_{(water)}$ < value$_{(water + remifentanil)}$). In addition, recording sessions in Experiment 2 occurred in the absence of remifentanil to avoid any confounding pharmacological effect of the drug. Based on the findings from Experiment 1, we predicted that within-neuron dopamine responses to drug and non-drug rewarded cues would be similar, even in the presence of behavioral drug-cue reactivity.

Rats in the opioid-exposed and opioid-naive groups were water restricted, and required to achieve >90% accuracy on both trial types A and B before proceeding to recording sessions (Fig. 4c, d). These occurred in a distinct operant chamber where no IV line was attached but a high level of accuracy was generally maintained for both trial types in both groups (Fig. 4f and Supplementary Fig. 5). Our data show enhanced reactivity to the drug cue in opioid-exposed rats as suggested by multiple lines of evidence. In opioid-exposed animals – but not opioid-naive – the relative trial accuracy index (% correct Cue B responses/% correct Cue A responses) across sessions was biased in favor of Cue B (Fig. 4g). Only port entry following cue termination determined correct vs. incorrect trial classification, so we examined port entries, in the correct trials, during auditory cue presentation. In opioid-exposed rats, the first feeder entry following Cue B onset was more likely to be congruent with the correct feeder entry compared to

the first feeder entry after Cue A in the same group and compared to the first feeder entry after Cue B in the opioid-naive group (Fig. 4h). We recorded the cumulative number of congruent port entries for both cue types in both groups and observed that accumulation of correct entries proceeded at a substantially faster rate for Cue B in opioid-exposed than in all three other conditions, detectable as both greater cumulative entries up to one second following cue termination (Fig. 4i) and greater slope during cue presentation (Fig. 4j). Behavioral reactivity was also seen in the more expedient lever-pressing response following lever extension in the opioid-exposed group. Instantaneous probability of a lever press (only one lever press was observed per trial, as the lever was immediately retracted) after lever extension, was quantified as hazard rate[35]. This was significantly higher in opioid-exposed vs. opioid-naive rats over the first two seconds following lever extension (Fig. 4k), resulting in a shorter time to 90% cumulative probability of lever press in the opioid-exposed (-2 s) vs. opioid-naive group (-6 s). These results demonstrate both selective drug-cue reactivity as well as generalized increase in motivation in the drug-exposed group.

While we observed behavioral drug-cue reactivity overall for the opioid-exposed group, one rat (rat 8) was a notable exception to this trend. This rat did not display drug-cue reactivity and showed more preference to the non-drug cue across all behavioral measures (Fig. 4l–o), even though it completed a similar number of trials and earned similar number of rewards to other opioid-exposed and opioid-naive rats (Supplementary Fig. 5e, f). The behavioral profile of rat 8 was more like the opioid-naive than the rest of the opioid-exposed rats (Fig. 4p). Thus rat 8 could serve as a case-control of opioid exposure for the role of dopamine firing in drug-cue reactivity.

## Dopamine firing responses to operant cues

With behavioral evidence of drug-cue reactivity, we then examined whether this behavior would be reflected in the activity of midbrain dopamine neurons. Dopamine neuron identification (*n* = 69) was like Experiment 1 (Supplementary Fig. 6). We performed our analyses both excluding units recorded from rat 8 due to the absence of behavioral cue reactivity (Fig. 5), and including the entire dataset (Supplementary Fig. 7).

Lever extension resulted in the largest dopaminergic firing response compared to other trial events, and was significantly higher in the opioid-exposed (*n* = 31, *N* = 5) compared to opioid-naive rats (*n* = 38, *N* = 5) (Fig. 5a–e). The discriminative auditory cues provoked significant phasic dopamine responses across all trial types. Although Cue C signaled the end of the trial and absence of reward, it resulted in a positive dopamine response comparable to rewarded cues in both opioid-exposed and naive groups, despite evidence of behavioral discrimination of the cue's identity (Fig. S5g). The response to Cue B was also slightly elevated relative to Cue A in both groups; however, the firing responses to both Cue A and Cue B were larger in the

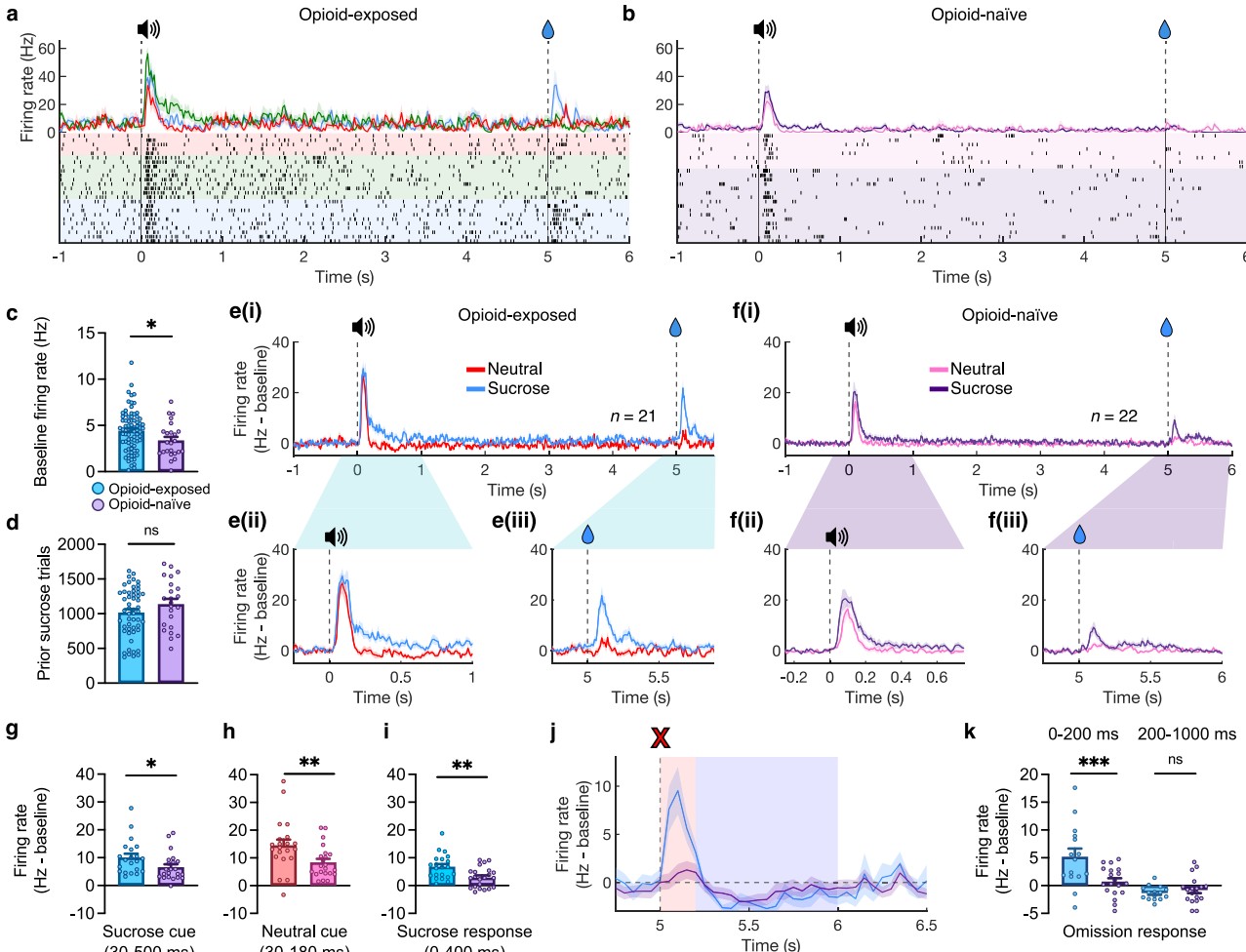

**Fig. 3 | Sensitized dopamine neuron responses in opioid-exposed rats in the Pavlovian procedure. a** PSTH and raster plot examples from a single opioid-exposed unit. Vertical lines indicate cue onset (0 s) and cue offset/sucrose delivery (5 s). Red = neutral trials, green = remifentanil (4 μg/kg), and blue = sucrose (40 μL). All traces show mean ± SEM[77]. **b** PSTH and raster plot examples from a single opioid-naïve unit. Display similar to *a*, with purple = sucrose and pink = neutral trials. **c** Mean firing rates prior to the start of session for each putative opioid-exposed (blue, *N* = 5 rats) vs. opioid-naïve (purple, *N* = 4 rats) units (two-tailed unpaired *t*-test, *t*(97) = 2.060, *n* = 99, *p* = 0.0421). All bars show mean + SEM. **d** Comparison of previous training quantified as number of sucrose trials prior to the recording session (two-tailed Mann-Whitney test, *n* = 75, *U* = 499, *p* = 0.2586). **e** (i) Average dopamine firing during sucrose and neutral (red) trials in opioid-exposed units (same data as Fig. 2). (ii) Close-up of cue response in opioid-exposed units. (iii) Close-up of reward delivery response in opioid-exposed units. **f** (i) Average dopamine firing during sucrose and neutral (red) trials in opioid-naïve units. (ii) Close-up of cue response in opioid-naïve units. (iii) Close-up of reward

delivery response in opioid-naive units. **g** Comparison of mean cue response to sucrose cues in opioid-exposed (blue) vs. opioid-naïve (purple) units two-tailed Mann-Whitney test, *n* = 43, *U* = 131, *p* = 0.0145). **h** Comparison of mean cue response to neutral cues in opioid-exposed (red) vs. opioid-naïve (pink) units (two-tailed Mann-Whitney test, *n* = 43, *U* = 120, *p* = 0.0064). **i** Comparison of mean response to sucrose delivery (0 – 400 ms after cue termination) (two-tailed Mann-Whitney test, *n* = 43, *U* = 118, *p* = 0.0054). **j** Trace of dopamine responses at time of reward omission in opioid-exposed (blue) and opioid-naïve units (purple). Red and blue shaded regions reflect positive (0-200 ms) and negative (200-1000 ms) response periods 1 and 2, respectively. **k** Comparison of omission firing responses during period 1 and 2 between opioid-exposed and opioid-naïve units (2-way RM ANOVA, Period factor: *F*(1,33) = 25.05, *p* < 0.0001; Exposure factor: *F*(1,33) = 6.087, *p* = 0.0190; Period x Exposure interaction *F*(1,33) = 9.022, *p* = 0.0051; Fisher's LSD test exposed vs. naive P1 *p* = 0.0002, exposed vs. naive P2 *p* = 0.6376.). Source data are provided as a Source Data file.

opioid-exposed than opioid-naive rats (Fig. 5g). After termination of the auditory cue and correct feeder entry, the light cue that confirmed correct responding and signaled upcoming reward delivery also resulted in significant dopamine activity. In opioid-exposed rats, this light-cue response was significantly greater than in the opioid-naive comparison group and there was no within-neuron difference in the mean response between the light cues predicting water vs. water + remifentanil/saline within each group (Fig. 5h). Overall, these results show that dopamine firing responses to cues associated with a (water + drug) reward were similar to water-only reward in the opioid-exposed group, and both were higher than opioid-naive group.

Finally, when the water reward was delivered, the phasic dopamine response was also greater for neurons in opioid-exposed than

opioid-naive rats (Fig. 5i). There was no such difference between the trial types A and B in either opioid-exposed or opioid-naive rats (Fig. 5i). No negative reward prediction errors were observed for the omitted second water reward in either opioid-naive or exposed groups (Supplementary Fig. 5h and i), likely due to its predictability. Of note, baseline firing measured during intertrial intervals was again significantly higher in dopamine neurons from opioid-exposed rats, like in experiment 1 (Fig. 5f).

## Relationship of firing to cue reactivity
We compared the activity of dopamine neurons from rat 8, which did not show cue reactivity despite an extensive history of remifentanil self-administration (achieved >90% accuracy during training sessions),

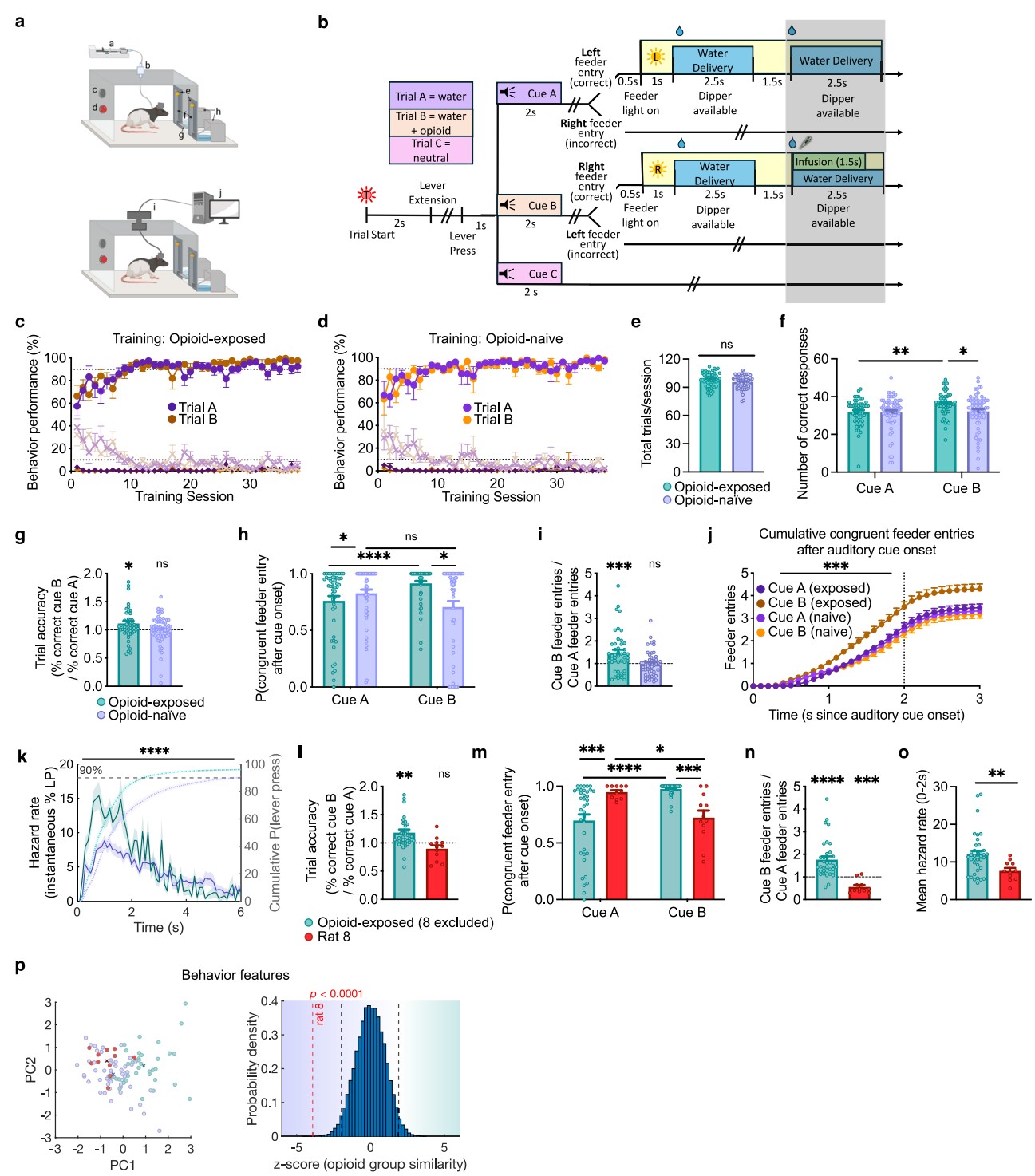

to the rest of the opioid-exposed population. Rat 8's neurons consistently showed significantly lower firing across all trial events, and its overall neural response profile was more like the opioid-naïve than the rest of the opioid-exposed units (Fig. 5j, k). These results are consistent with a role of enhanced phasic dopamine responses in cue reactivity. The first PC (PC1) was extracted from the cue reactivity and dopamine response analyses in Fig. 4p, k, respectively. We treated the PC1 from the respective analyses as a compounded approximate measure of behavioral cue reactivity and enhanced dopamine responsiveness: In both analyses, PCs 1 and 2 combined accounted for ~85% of the data's variance, of which >50% was captured by PC1 alone. Further, in the behavioral analysis, all included variables were designed to correspond

positively with reactivity and loaded positively on PC1. Finally, in both analyses, the opioid-exposed group was largely higher in PC1 than the opioid-naïve group, consistent with our direct comparisons of the underlying individual variables. In support of the correlative relationship between dopamine responses and behavioral drug-cue reactivity, we found a significant moderate, positive correlation between the first principal components of neuronal firing and behavioral cue reactivity from the sessions in which they were recorded (Fig. 5l). The difference in dopamine firing between the opioid-exposed and naïve groups or rat 8 was not reflected in the number of trials completed or rewards received (Supplementary Fig. 5e, f). We also correlated the first principal components for behavior and dopamine firing in only opioid-

**Fig. 4 | Operant experimental procedure demonstrates behavioral drug-cue reactivity. a** Schematic of experimental setup during training (top) and recording (bottom) sessions in Experiment 2. Boxes equipped with infusion pump (**a**), swivel (**b**), speaker (**c**), trial light (**d**), feeder lights I, feeders (**f**), retractable lever (**g**), dippers (**h**), commutator (**i**), and recording computer (**j**). **b** Experimental design. Gray shading indicates omitted reward in block two of recording trials. **c** Behavior training results for opioid-exposed animals ($n = 5$) = 5). Circles indicate % correct responses, Xs indicate % incorrect, and diamonds indicate % not responded to. Error bars = SEM. **d** Similar to 4c for opioid-naive rats ($n = 6$). **e** Total number of trials completed per recording session for opioid-exposed and opioid-naive rats (All bars = mean + SEM $n = 106$ sessions from $N = 11$ rats, two-tailed Mann-Whitney test, $U = 1135$, $n = 106$ sessions from $N = 113$ sessions from $N = 11$ rats, = 11 rats, $p = 0.0975$). **f** Number of correct responses per session to Cue A and Cue B (2-way RM ANOVA, Cue factor: $F_{(1,114)} = 4.786$, $n = 113$ sessions from $N = 11$ rats, $p = 0.0307$. **g** Trial accuracy response index for opioid-exposed and opioid-naive animals (two-tailed one-sample $t$ tests, $H_0$: $\mu = 1$. Opioid-exposed: $t_{(64, 47)} = 2.168$, $n = 48$, $p = 0.0353$; Opioid-naive: $t_{(63)} = 0.8954$, $n = 64$, $p = 0.3740$). **h** Probability that first feeder entry after cue onset is congruent with cue identity (2-way RM ANOVA, opioid exposure factor: $n = 100$ sessions from $Nn = 47$, = 11 rats, $F_{(1,101)} = 5.032$, $p = 0.027$; interaction $F_{(1,101)} = 8.621$, $p = 0.0041$). **i** Ratio of total congruent feeder entries for Cue B/Cue A in both groups (two-tailed one-sample $t$ tests. Opioid-exposed: $t_{(46)} = 3.343$, $n = 47$, $p = 0.0017$; Opioid-naive: $t_{(5} = 110$ sessions from $N = 11$ rats, 4) $= 0.6371$, $n = 55$, $p = 0.5268$). **j** Cumulative congruent feeder entries following auditory cue onset (F-test for difference of slopes, $F_{(3,156)} = 5.482$, $n = 10$ $p = 0.0023$; rat 8: $t2$ sessions from $= 1.730$, $N = 11$ rats, $p = 0.0013$). **k** Left axis: Session-mean of lever press hazard rate. 2-way Mixed effects model, Opioid-exposure factor: $F_{(1,108)} = 24.04$, $n = 110$ sessions from $N = 11$ rats, $n = 49$ sessions from $N = p < 0.0001$). Right axis: cumulative % of sessions with lever press at time $t$ is shifted to the left in opioid-exposed rats. All traces show mean ± SEM. **l–n** As in **g–i** respectively, but for rat 8 (red) and all other opioid-exposed animals: l (Opioid-exposed: $t_{(36)} = 3.284$, $n = 37$, $p = 0.0023$; rat 8: $t_{(11)} = 1.730$, $n = 12$, $p = 0.1115$); m (interaction $F_{(1,47)} = 24.4$, $n = 49$ sessions from $N = 5$ rats, $p = <0.0001$); n (Opioid-exposed: $t_{(34)} = 5.352$, $n = 35$, $p < 0.0001$; rat 8: $t_{(10)} = 5.000$, $\boldsymbol{n} = 11$, $p = 0.0005$). **o** Mean hazard rate for lever press during 0-2 s after lever extension in rat 8 vs. other opioid-exposed rats (two-tailed Mann-Whitney test, $n = 49$, $U = 91$, $p = 0.0038$). **p** Left: First two principal components (PCs) extracted from composite behavior measures for opioid-naive rats (purple), rat 8 (red filled), and all other opioid-exposed rats (teal). Crosses represent corresponding centroids. Right: Mean difference of rat 8 centroid under true labels (red dashed line) compared to bootstrapped distribution under shuffled labeling shows greater similarity of rat 8 to the opioid-naive group. Black dashed lines indicate two-tailed 95% threshold of null distribution. Source data are provided as a Source Data file.

exposed rats, excluding the opioid-naive group. We found a positive correlation (Spearman's $\rho = 0.367$, $n = 22$, $t_{(20)} = 1.768$) that approached our significance threshold ($p = 0.059$).

## Discussion

This study challenges the prevailing hypothesis of cue reactivity that opioids act as noncompensable positive reward prediction errors, driving selective enhancement of dopamine response to drug-related cues. We found a non-selective effect of opioid exposure on dopamine neuron activity, particularly elevated baseline firing rate, and enhanced phasic responding across cue and reward types and in multiple distinct experimental setups. There was largely no difference in the magnitude of dopamine firing responses to cues predicting an opioid drug compared to a natural reward, even when the drug reward was clearly of higher value (drug + water vs. water). These findings contradict the widely held reward prediction error-based assumptions of abnormal cue reactivity, which suggest that dopamine neurons systematically encode lower values for natural reward cues than drug cues, or even natural reward cues in non-drug-exposed individuals[5,6].

Although we observed a general sensitization across different cue types, this sensitization was not entirely uniform. For instance, the dopamine firing response remained graded by trial type (rewarded vs. unrewarded). This was observed in the delayed valuation phase of the dopamine response to neutral cues and sucrose omission trials where the time course of the dopamine response differed between rewarded and unrewarded trials. Schultz et al. proposed a model for phasic dopamine response in which two components signal salience, then reward value[32]. In the data from our Pavlovian study, remifentanil, sucrose, and neutral cues all showed a similar positive dopamine firing response during the initial salience-detection period (30–180 ms) in opioid-exposed rats (but not in opioid-naive rats), suggesting a state of general hypervigilance, at least towards the sensory modality of reward predictive cues. However, during the valuation period (180-500 ms), responses to the neutral cue rapidly dropped, while firing rates stayed above baseline for both remifentanil and sucrose cues. A similar detection/valuation firing discrimination was observed in sucrose omission trials: a detection phase immediately after cue offset showed greater firing response in opioid-exposed compared to opioid-naive; however, a valuation phase around expected reward delivery time showed equivalent negative reward prediction errors between groups. Thus, despite their increased excitability, these neurons are still able to signal value-relevant disappointment, as both neutral cue and reward omission predict a longer delay to the next reward. In the operant experiment, there is also evidence for

distinct detection/valuation phases in the response to the auditory cue, which is graded according to reward value. Interestingly, however, the detection phase of this cue response is not enhanced in the opioid-exposed group relative to opioid naive, unlike every other cue examined within the same subjects in the same experiment. One possibility could be that the auditory cue was discriminative and hence possibly involved different regulatory processes of dopamine firing (e.g., attentional top-down control). Another possibility is that opioid-exposed animals experience a steeper temporal discounting factor, resulting in greatly magnified response of cues near reward (light cue) and less enhancement earlier on (auditory cue). However, the enhancement of the dopaminergic response to the lever extension at the beginning of each trial goes against an explanation in terms of harsher discounting.

An alternative to the noncompensable reward prediction error hypothesis of cue reactivity that our findings support is that exposure to opioids, rather than accumulating value on drug-specific cues (cached value) and thus selectively enhancing dopaminergic responses to these cues, results in increased excitability of midbrain dopamine neurons, which manifests as a generalized reinforcement gain. This gain causes enhanced but reward-dependent dopamine response to all salient cues that occur in the same spatiotemporal context as drug use including non-drug predicting and neutral cues. Such non-specific cue reactivity has been shown clinically in smokers, where reactivity to drug and non-drug cues covary significantly[36], and in Parkinson's disease patients who develop impulse control disorders after treatment with dopamine agonists where natural reward cues result in greater dopamine release in patients with impulse control disorders compared to those without[37].

One possibility for such an effect is that drug-induced dopamine release is sustained over minutes, unlike the physiologic brief and discrete dopamine release in response to natural rewards, and hence causes long-lasting changes in signaling cascades and gene expression that outlast the time course of dopamine release. These changes, which normally reinforce drug-related cues and behaviors, overlap with and thus also reinforce other spatiotemporally overlapping cue-reward associations in the animal's environment, effectively causing a generalized increase in the gain on future reward expectation in the reward prediction error calculation. Clinical and pre-clinical imaging studies of various substance use disorders show increased mesolimbic dopamine release in response to drugs or drug-associated cues, indicative of a sensitized hyperdopaminergic state[38–42], but other studies suggest that striatal dopamine transmission is blunted in drug-experienced subjects[43,44]. These disparate findings can be accounted

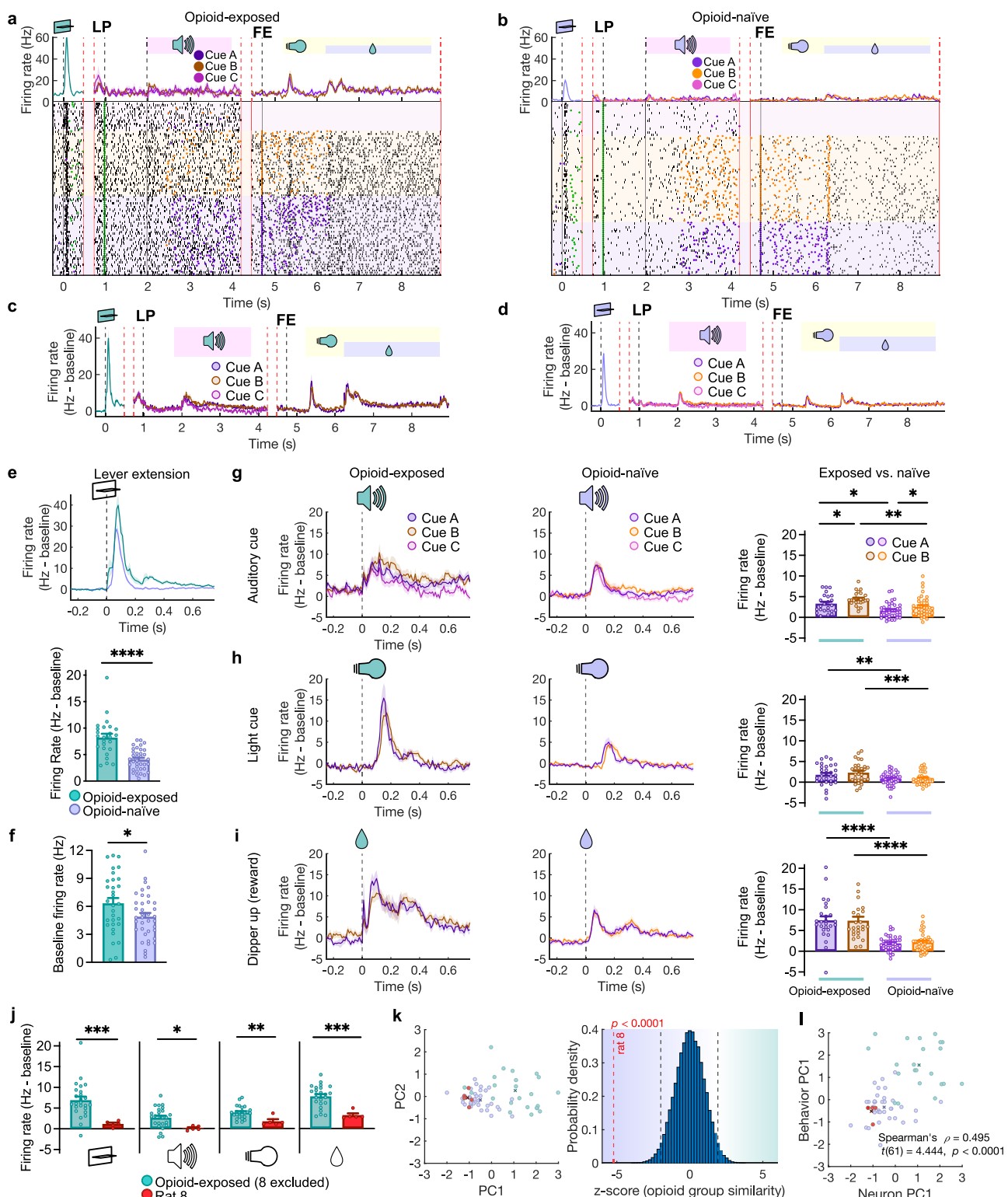

for by considering the time interval since the last drug exposure, context of drug administration or cue presentation, and drug exposure procedure (intermittent vs. continuous)[39,40,42]. Because imaging often occurs in a clinical context which is not associated with drug availability and at a time of maximal tolerance, these results may reflect a tolerant state of the dopaminergic system rather than a more dominant dopaminergic sensitization state in drug-associated context (where relapse is most likely) that is most pronounced during drug use or in early abstinence. Alternatively, dopamine release can be heavily influenced by downstream synaptic factors like inputs from

cholinergic interneurons onto dopamine axon terminals in the striatum, so striatal dopamine levels may differ substantially from somatic VTA dopamine firing rate[45]. A further difficulty that arises in comparing these data to the clinical cue reactivity literature is that many cue reactivity procedures reference their results against neutral stimuli, and do not directly compare drug stimuli to other motivationally relevant natural-reward related stimuli[46]. Such studies have rarely been carried out alongside neuroimaging in opioid-use disorder. Further, in addition to the well-established effects of acute opioids on increasing dopamine neuronal firing[47–49], opioid use is linked to long-lasting

**Fig. 5 | Sensitized dopamine responses to drug and non-drug cues in opioid-exposed rats with behavioral drug-cue reactivity. a** PSTH and raster plot examples from a single opioid-exposed unit. Vertical red lines indicate timeline discontinuity due to variable intervals within trials. Vertical black lines indicated discrete events: lever extension, lever press, and rewarded feeder entry. Purple trace includes all trials, which subsequently split based on trial identity to corresponding colors. Purple and orange dots indicate left and right feeder entries respectively. Green dots indicate lever presses. All traces show mean ± SEM. **b** Similar to a but for an opioid-naive unit. **c** Mean PSTH from opioid-exposed units ($n = 31$, $N = 5$ rats) displayed like in (**a**). **d** Mean PSTH from opioid-naive units ($n = 38$, $N = 6$ rats). **e** Top: PSTH of baseline-subtracted firing rate around lever extension for opioid-exposed and opioid-naive units. Bottom: mean firing rate for both groups (two-tailed Mann-Whitney test, $n = 62$, $U = 124$, $p < 0.0001$). All bars show mean + SEM. **f** Baseline firing rate for opioid-exposed (teal) and opioid-naive (purple) units (two-tailed unpaired $t$-test, $t(67) = 2.117$, $n = 69$, $p = 0.0379$). **g** Responses of opioid-exposed and opioid-naive units to the auditory cues. Right: mean firing rates (2-way RM mixed-effects analysis, $n = 67$, Cue factor: $F(1,64) = 10.79$, $p = 0.0017$; Exposure factor: $F(1,65) = 8.157$, $p = 0.0058$, Fisher's LSD exposed A vs exposed B $p = 0.0226$, exposed A vs naive A $p = 0.0170$, exposed B vs naive B $p = 0.0096$, naive A vs naive B $p = 0.0237$). **h** Responses of opioid-exposed and opioid-naive units to light cue. Right: mean firing rates (2-way RM mixed-effects analysis, $n = 69$, trial-type factor:

$F(1,60) = 2.110$, $p = 0.1515$; Exposure factor: $F(1,62) = 11.96$, $p = 0.0010$, Fisher's LSD exposed A vs exposed B $p = 0.0571$, exposed A vs naive A $p = 0.0070$, exposed B vs naive B $p = 0.0002$, naive A vs naive B $p = 0.9430$). **i** Responses of opioid-exposed and opioid-naive neurons to water reward delivery. Right: mean firing rates (2-way RM mixed-effects analysis, $n = 69$, trial-type factor: $F(1,61) = 0.1357$, $p = 0.7139$; Exposure factor: $F(1,62) = 38.41$, $p < 0.0001$, Fisher's LSD exposed A vs exposed B $p = 0.8570$, exposed A vs naive A $p < 0.0001$, exposed B vs naive B $p < 0.0001$, naive A vs naive B $p = 0.7218$). **j** Mean responses of neurons from rat 8 (red) vs. all other opioid-exposed units in response to lever extension ($t(27) = 3.695$, $n = 29$, $p = 0.0010$), auditory cue (A and B; $t(27) = 2.915$, $n = 29$, $p = 0.0071$), light cue (A and B; $t(28) = 2.209$, $n = 30$, $p = 0.0355$) and reward delivery (A and B; $n = 30$, $U = 8$, $p = 0.0008$). **k** Left: First two principal components (PCs) extracted from composite firing measures for opioid-naive rats (purple), rat 8 (red filled), and all other opioid-exposed rats (teal). Crosses represent corresponding centroids. Right: Mean difference of rat 8 centroid under true labels compared to bootstrapped distribution under shuffled labeling plotted like in Fig. 4p shows greater similarity of rat 8 to the opioid-naive group. **l** Scatter plot of principal component 1 from neuron firing activity analysis in K vs. principal component 1 from behavior analysis from Fig. 4p shows moderate positive correlation (two-tailed $t$-test for Ho: $\rho = 0$) between behavioral cue reactivity and dopamine firing response. Source data are provided as a Source Data file.

increased excitatory and reduced inhibitory synaptic transmission onto midbrain dopamine neurons during abstinence[50,51], which could contribute to increased dopamine neuronal excitability. The dopamine-dependent psychomotor sensitization after opioids[52] and increased dopamine neuronal firing to acute opioids after chronic opioid exposure[38] also point towards increased excitability of dopamine neurons. This is also supported by the observed increase in baseline firing rate in our opioid groups. Tonic firing has been theorized to reflect real-time value estimates[29,53], which concords with a general motivational sensitization reflected in the increased hazard rate of lever pressing (note, however that we did not find a significant relationship between individual neurons' baseline firing rates and session hazard rate). In addition, the generalized reinforcement gain model is more consistent than the noncompensable reward prediction error model with prior studies that failed to show overwhelming drug preference in animal models. In free-choice tasks, animals often exhibit strong (but dose-dependent[54]) preferences for even modest food rewards over drug rewards[55,56]. On the reward prediction error model, this is difficult to explain, as the runaway overvaluation of the drug state ought in the long run to overwhelm any other state, no matter how rewarding, resulting in almost exclusive preference for drug. If, however, opioid use resulted in nonselective augmentation of dopamine activity (as our data show) that drives valuation of all reward cues, then an impulsive rat faced with the choice of immediate food reward and pharmacokinetically delayed drug reward could easily choose the former. This alternative model of non-selective reinforcement gain through sensitization of dopamine function is also consistent with prior literature showing that activation of dopamine circuitry by drugs or medications heightens sensitivity to rewards that are not directly predictive of receipt of the pharmacological agent. For example, cue-induced responding for natural rewards can be increased by exposure to opioids, amphetamines, or cocaine[57–60] including in a drug-free state and in a distinct context[60]. Similarly, patients taking dopamine-augmenting medications for movement disorders can develop addiction-like compulsive behavioral problems as a side effect[61,62].

Because our findings challenge the straightforward reward prediction error interpretation of drug-induced dopamine signaling, they are also consistent with non-reward prediction error models. For instance, the ANCCR model posits that phasic dopamine marks events for learning causal relationships, resulting in dopamine responses to reward-predictive cues resembling classical reward prediction errors in many contexts[20]. In this model, if the innate "importance" of reward indicated by phasic dopamine were enhanced by the presence of drug,

the learned "importance" of the predictive cues would also be enhanced but can stabilize without driving subsequent dopamine responses to zero. Indeed, rats working for optogenetic stimulation of VTA dopamine neurons achieve stable, non-maximal levels of responding despite persistent stimulated dopamine release[63]. Thus, neither opioids nor optogenetically elicited phasic dopamine supports runaway overvaluation as the noncompensible reward prediction error model suggests.

Our results in Experiment 2 show a disconnect, in the opioid-exposed group, between the cached value of reward cues measured with dopamine firing (equal dopamine neuron firing) and their actual value reflected in the drug-cue reactivity (unequal behavior). However, our data also show a positive relationship between aggregated measures of enhanced dopamine firing and drug-cue reactivity. Thus, although our data rule out the possibility that this behavioral selectivity is generated by a difference in dopamine firing to drug vs. non-drug cues, it remains unknown how much enhanced excitability of dopamine neurons contributes to reactivity. Compelling cross-species evidence suggests that mammalian reward learning balances elements of model-free (e.g., temporal-difference) and model-based learning[64,65]. It is possible that the non-specific dopamine enhancement we observe here could promote behavioral activation which is sculpted to promote drug-seeking by causal information supplied by structures implicated in model-based learning, such as the orbitofrontal cortex. A recent report has suggested a similar role for the orbitofrontal cortex in promoting drug addiction by interfering with natural reward consumption[66]. It is also possible that behavioral selectivity is dictated by regulation of dopamine release distally at axon terminals, which cannot be directly inferred from spiking activity alone, and may be regulated independently from somatic firing[67]. Dopamine dynamics that are distally regulated in this manner would not be observable by our approach of somatic electrophysiology and would require direct measurement of extracellular dopamine (e.g., by fiber photometry) to clarify. However, other reports have shown that closer matching of somatic and axonal dopamine dynamics substantially alleviates the somatic-axonal discrepancy[68].

In this study, we used only two female rats in Experiment 1, and all male rats otherwise. The experiment was not powered to detect a sex difference in dopamine firing between males and females, so we cannot rule out the possibility that including a larger sample of females would reveal differing patterns of response to opioid exposure that could be relevant for understanding sex differences in opioid use and susceptibility to addiction. A further limitation is that cues in both experiments were not counterbalanced, so differences in sensory properties may contribute to the effects (and absence of predicted

effects) we report. While the main findings of experiments 1 and 2 were consistent despite using completely different sensory cues (pure auditory tones vs. lights and melodic cues), the possibility of such sensory confounding cannot be eliminated based on these data. Finally, our experiments were not designed to test the causal role of enhanced dopamine firing on behavior, and though the observed correlation between dopamine activity and motivated behavior in Experiment 2 is suggestive and concordant with clinical observations of increased dopamine release to natural reward cues in patients with impulse control disorders[37], its causal significance is as yet unproven.

In our data, drug and non-drug cues were learned in a single context, so it is unclear whether the enhanced dopamine response to non-drug cues and rewards is spatiotemporally-specific. Because we hypothesize that the observed enhancement is a result of aberrant learning around drug use, we further postulate that our findings are drug-context specific and thus carry important implications for the clinical setting, where patients frequently relapse shortly after discharge to their home environment where they experienced drug use[69]. This framework is supported by improved clinical outcomes and high abstinence rates in substance use disorders patients who are discharged into new environments, not associated with the previous drug context[70]. In summary, results from this study warrant revised models of dopaminergic function following sustained drug use and provide insights into the neurobiology of cue reactivity which is critical to relapse prevention.

## Methods

### Subjects
All experimental procedures were conducted in accordance with the guidelines of the National Institutes of Health Guide for the Care and use of Laboratory Animals and approved by the Animal Care and Use Committee of the University of Pittsburgh (Experiment 1) and the National Institute on Drug Abuse (Experiment 2) (protocols 21028545 and 20-CNRB-108). Experiment 1 involved 11 adult Long-Evans rats (age 9–15 weeks, mean = 10.5 weeks, 9 males, 2 females) with 7 rats in the opioid-exposed group and 4 rats in the opioid-naive group. Experiment 2 involved 11 male Long-Evans rats (age 9–12 weeks) with 5 rats in the opioid-exposed group and 6 rats in the opioid-naive group. All rats were obtained from Charles River Laboratories. Rats were single-housed. Following recovery from all surgical procedures, rats were restricted to roughly 1 h of water access daily following behavior to maintain at least 80% of baseline weight and *ad libitum* food access.

### Electrode surgeries
For Experiment 1, rats were implanted with 32-channel drivable electrode bundles targeting left VTA (ML = −0.50 mm, AP = −5.40 mm, DV = −7.40 mm), protected by plastic caps. Electrodes were constructed from 25 μm formvar-insulated NiChrome wires (A-M Systems Carlsborg, WA)[71]. The wires were bundled in two 27-gauge cannulas centered 740 um apart and mounted in a custom 3D-printed microdrive. Prior to implantation, wires were trimmed to 1–2 mm, spread to allow ≥ 25 μm between them, and gold-plated to an impedance of 400–700 kΩ (at 100 Hz). Surgeries were performed under isoflurane with aseptic technique, and 0.5 mg/kg Carprofen was provided for two days for pain management. For two weeks, oral cephalexin and topical Neosporin were also provided.

For Experiment 2, rats were implanted with similar electrodes targeting the VTA. In this experiment, most rats had 8-channel electrodes, while one had 32 channels.

### Catheter surgeries
For Experiment 1, after two weeks of recovery from electrode surgery, indwelling intravenous catheters were implanted in the right jugular vein and tunneled to a vascular access port on the rat's back (Instech) as previously described[72], followed by a third week for recovery, in which the catheter was flushed daily with 0.9% saline and gentamicin (4.25 mg/kg). Thereafter, catheters were flushed daily with a solution of saline, heparin and enrofloxacin, and tested weekly with propofol to ensure patency.

For Experiment 2, IV catheter implantation was done as in Experiment 1. Both cohorts underwent similar long-term treatment with saline and antibiotics, and regular propofol testing to ensure maintained patency.

### Pavlovian procedure in Experiment 1
Rats were water restricted (30–60 min access following behavior each day) and trained in a customized operant chamber equipped with an overhead tether for IV fluid delivery, a well connected to solenoid valves for delivery of oral sucrose (10% w/v) and vacuum for removal, and a speaker opposite the port to play auditory cues. During training, the beginnings and ends of sessions were marked by a house light turning on and off respectively. Remifentanil was prepared each day from stock stored at −20 °F.

Training consisted of three trial types with distinct 5 s auditory cues. In sucrose trials, a 5 s 8 kHz tone was immediately followed by sucrose dispensed at the well with a vacuum activated later to remove any unconsumed liquid between trials. In remifentanil trials, a noiseless IV pump dispensing remifentanil was activated simultaneously with onset of a 5 s 12 kHz tone to minimize the delay between cue onset and onset of pharmacological effect of remifentanil. In neutral trials, a 5 s cue rapidly cycling between 2.5, 3, 3.5, and 4 kHz was followed by no consequence. Intertrial intervals between trials were selected pseudorandomly and lasted 11-150 s, (mean 68.7 s or 69.9 s, following sucrose and neutral trials) or 110-320 s (mean 173 s, following remifentanil trials). Sucrose and remifentanil rewards (sucrose from solenoid and remifentanil IV pump) were omitted in a randomly selected 10% of trials. Training sessions consisted of one or two blocks per session with 25–50 trials per training block (typically 2:2:1 ratio of remifentanil, sucrose, and neutral trials) with occasional sucrose or remifentanil reward omission (~10% of trials). Opioid-naive rats were trained only on sucrose and neutral cues.

### Progressive ratio testing
Six male Long-Evans rats trained in the Pavlovian procedure were subsequently trained on a progressive ratio task. Training began with an FR1 schedule for 4 μg/kg IV remifentanil reward paired with the familiar drug cue per lever press. Each session lasted one hour, and training continued for at least five sessions, and until each rat received at least 50 infusions in one session. Progressive ratio testing was carried out with exponentially escalating response requirements[73] (steps = 1, 2, 4, 6, 9, 12, 15, 20, 25, 32, 40, 50, 62, 77, 95, 118, 145, 178, 219, 268, 328, 402, 492, 603, and 737). Each session was terminated after three hours or a period of 30 min with no reward. Recorded scores are the highest of ~3 PR sessions per animal. After remifentanil testing, rats were trained to press a lever located on the other side of the operant chamber for 40 μL oral sucrose. Training began in FR1, with each lever press resulting in a familiar sucrose cue, and delivery of liquid at the same location as in Pavlovian conditioning. Once again, training was maintained for at least five sessions, or until at least fifty rewards were collected in a one-hour session. Subsequently, progressive ratio testing for sucrose reward was carried out according to the same schedule, and recorded scores represent the highest of ~3 sessions per animal.

### Operant procedure in Experiment 2
In Experiment 2, rats were water restricted and trained with trials resulting in natural reward, (natural+drug) reward, and no reward. Natural reward consisted of water, and the drug reward (4 μg/kg IV remifentanil) was paired with an identical water reward. All operant trials began identically with illumination of an overhead trial light, followed two seconds later by extension of the lever. A lever press

resulted in lever retraction and, one second later, by one of three discriminatory melodic auditory cues played to indicate the trial type. Note that due to the proximity of lever presses to lever extension, firing responses during lever press are difficult to compare, as neural responses do not return to baseline before the behavioral event (Supplementary Fig. 5d). The auditory cues were pseudorandomized and signaled the location and identity of the reward (Cue A, "siren": water only, left port; Cue B "white noise": water + remifentanil/saline, right port; Cue C "3 kHz beeping 2x/s": no reward - neutral trial). A port entry during cue presentation was inconsequential irrespective of congruency with correct feeder side. A correct port entry after the offset of Cue A resulted 0.5 s later in the illumination of a light cue within the feeder port for the whole duration of reward delivery. One second after the light cue was turned on, two water rewards were delivered via a motorized dipper (Coulbourn H14-05); each water reward was 40 µl and was delivered over 2.5 s with 1.5 s interval between the rewards during which the dipper was retracted outside the feeder port (Fig. 4a, b). A correct port entry after the offset of Cue B resulted in a similar cascade of cues and rewards as to Cue A response (including 2 ×40 µl water rewards) but with the additional activation of an infusion pump to deliver IV remifentanil (4 µg/Kg/50 µl infusion) or saline at the onset of the second water reward delivery (Fig. 4b). The intertrial interval for correct trials A and B types was 60 ± 15 s. Incorrect response to Cue A or B resulted in a prolonged intertrial interval (120 ± 15 s). If rats failed to respond to Cue A or B within 10 s, a new trial was initiated and the lever re-extended. Type C cues were inconsequential, and Cue C offset was followed by a shortened intertrial interval (30 ± 10 s). Each training session was 4 hrs. Type A, B, and C cue trials were pseudorandomized from a list with typically 1:1:1 ratio. Refer to Fig. 4a, b for detailed schematic of recording chamber arrangement.

## Single unit recordings and spike sorting

During recording, electrodes were connected to a headstage tether which was prevented from tangling with the IV tether by a custom motorized commutator in Experiment 1 (Plexon, inc.). Single unit recordings were acquired with the Plexon OmniPlex system. Signals were digitized at 40 kHz and band-pass filtered at 0.1–7500 Hz, then digitally high pass filtered at 0.77 Hz. Spike data was separated from LFPs by an additional 150 Hz high-pass filter. Between sessions, electrodes were advanced 40-80 um. Spikes were isolated and single units sorted manually in Offline Sorter (Plexon, Inc.) with an SNR cutoff of 3:1 ($\sigma_{spikes}^2/\sigma_{noise}^2$).

## Recording sessions in the Pavlovian procedure (Experiment 1)

Single-unit recordings in Experiment 1 occurred throughout the behavioral training sessions described above.

## Recording sessions in the operant procedure (Experiment 2)

Single-unit recordings in Experiment 2 occurred in distinct sessions from the training sessions. Recording sessions were generally similar to the training sessions with the following exceptions: The recording sessions involved 2 blocks during which the IV infusion was omitted for both groups, and recordings occurred in a customized operant chamber in a different location than training boxes where no IV line was attached. Block one was identical to training except the lack of IV infusions and lasted until the rats correctly responded to twelve rewarded trials. Once this requirement was met, rats advanced to block two, which was identical to block one except for the omission of the second reward in rewarded trials. This design allows for the examination of negative reward prediction errors in opioid-exposed and opioid-naive groups. A recording session usually consisted of a maximum 72 total correct type A and B trials, or 90 min duration, with typically 2:2:1 ratio of types A, B, and C trials). The intertrial intervals were shorter than training sessions (correct response to Cue A/B: 35 ± 5 s; incorrect response to Cue A/B: 50 ± 10 s; Cue C: 20 ± 5 s).

Between recording sessions, rats were retrained for at least 2 sessions to prevent behavioral extinction and were required to achieve >90% accuracy on both trial types A and B before a new recording session was carried out.

## Dopamine unit identification

Waveform properties were calculated from the average waveform extracted from each unit. The amplitude ratio was defined as the ratio of the difference to the sum of the first positive peak detected before the first negative peak and the first negative peak. If the first peak was negative, (no preceding positive peak), the first positive peak was taken to have amplitude of 0. The half duration was defined as the time from the first negative peak to the next positive peak, or the end of the waveform (at 1.1 ms) if no such positive peak was detected[19,71,74]. After screening for cue responsiveness in Experiment 1, 225 units were considered for further analysis. To identify putative dopamine neurons, we used an activity-based clustering approach that identifies dopamine neurons based on canonical phasic responses to conditioned cues and reward, which has been shown to accurately discriminate genetically identified dopamine cells[13,26,27].

Each of the units' spikes was isolated in a period 0.5 s before and 5.5 s after each sucrose cue, and a normalized response was calculated with the auROC method. The first three principal components of these normalized responses were clustered hierarchically into three groups using Ward's distance metric (cluster 1 excited, n = 39, cluster 2 inhibited, n = 30, cluster 3 phasic, n = 165). The group that displayed classic phasic reward prediction error-like dopamine activity and was labeled as putative dopamine. Within the putative dopamine cluster, we observed a subpopulation with delayed peak firing, highly labile firing, and sustained inhibition to reward. We calculated peak firing time as the time of highest activity 50–1000 ms after sucrose cue onset, binned at 20 ms, firing variability as the coefficient of variation of ISIs recorded throughout the sessions excluding periods 0–500 ms after cue/reward events (where phasic activity is expected), and inhibition as $(R-B)/(R+B)$ where $R$ was the mean response 1000–2000 s after reward delivery and $B$ was the session baseline firing rate. We performed principal component analysis on normalized log(peak firing time), log(ISI CV), and inhibition and used $k$-means clustering on the first two principal components ($k = 2$, Euclidean distance, 20 replicates) to separate this group ($n = 21$) from canonical dopamine neurons. We excluded any units exhibiting baseline firing > 12 Hz from this clustering ($n = 39$). Finally, 6 units were removed manually (Supplementary Fig. 9a) leaving 99 putative dopamine neurons in our analysis. Since dopamine units were not directly genetically identified, we cannot rule out the possibility that some excluded units are dopamine units that deviate from canonical dopamine activity. However, if such neurons are present, they are not well-characterized in prior literature, and their activity is not predicted by conventional models of dopamine activity. Seventy-five of the putative dopamine neurons were recorded from rats with prior opioid exposure (opioid-exposed), while the remaining 24 were recorded from comparison rats which received no opioids (opioid-naive). There was no significant difference in the waveform properties between the opioid-exposed and opioid-naive groups (Supplementary Fig. 8a), and putative dopamine neurons exhibited classic phasic response to cues, with firing response to cues peaking at -100 ms (Fig. 2 and Supplementary Fig. 8b) in both groups. Further, there was no difference in the bursting behavior in terms of frequency of bursts, percentage of spikes in bursts, or distribution of burst sizes (Supplementary Fig. 8c–e).

Identification in Experiment 2 was achieved with a similar approach. Since we lacked a simple cue-reward pairing, a composite cue-reward period was created by appending activity recorded during around initial lever extension (-250 to 1000 ms) to activity around reward delivery (−250 to 500 ms). Hierarchical clustering on auROC

signals again yielded three groups exhibiting sustained excitation ($n = 55$), inhibition ($n = 39$), and phasic firing ($n = 187$). After eliminating those units in the 'phasic' cluster with high baseline firing rate ( > 12 Hz; $n = 75$), we again observed a subset of units with sustained inhibition. Peak firing time, ISI CV, and inhibition were calculated as in Experiment 1, except that the inhibition response was the lower of mean firing 0-1000 ms before or after reward delivery. Principal component analysis, as in Experiment 1, was followed by $k$-means clustering ($k = 2$, Euclidean distance, 20 replicates), eliminating 30 units. A further three units were eliminated manually (Supplementary Fig. 9 b) and the remaining units were classified as putative dopamine ($n = 69$).

## Rat 8 similarity analysis

To test the similarity of rat 8 behavior to the opioid-naive vs. opioid-exposed rats, we performed principal component analysis on four behavioral measures (cumulative feeder entry ratio, correct trial index, mean hazard rate, and congruent feeder entry ratio) and retained the first two principal components. We randomly sampled with replacement from the naive sessions and the opioid sessions (excluding those from rat 8) and computed the difference in Euclidean distances between these points and the centroid of rat 8's behavioral data (mean distance between opioid-naive sessions to centroid of rat 8 from mean distance between opioid-exposed sessions and centroid of rat 8). Averaging the differences in distances obtained by 2000 such samples allowed us to generate a bootstrapped estimate of the mean similarity to the naive and exposed groups with negative values reflecting greater similarity of rat 8 behavior to the naive group, while positive values reflecting greater similarity to the exposed group. We generated a null distribution by randomly relabeling the behavioral data from both groups (excluding rat 8) with the same proportion of exposed and naive sessions and computing 50,000 relabeled similarity scores (Fig. 4p). Statistical significance was assessed by computing the percentile of the true similarity score based on the null distribution.

To test for similarity of single unit activity, we performed principal component analysis on four measures of spiking activity (mean response to lever extension, auditory cues A and B, light cues A and B, and rewards A and B). Subsequent analysis on the first two principal components was performed identically to behavior data.

## Data processing

Peristimulus temporal histograms were constructed from raw spike trains with bins of 10 ms width, except where otherwise noted. Smoothing was carried out by convolving the binned data with a causal exponential kernel over 200 ms (Thompson et al, 1996). Lines and surrounding shaded areas in traces show mean ± SEM of data.

In Experiment 1, well entries and exits were recorded and used to calculate the PSTH of probability of presence in the sucrose well at times relative to analysis events. Similarly, in Experiment 2, lever presses were recorded to calculate the hazard rate and latency to press after lever extension, and feeder port entries and exits were recorded to calculate feeder entry probability and cumulative feeder entries during auditory cue presentation.

For most comparisons, except where otherwise noted, mean phasic firing was calculated as the mean firing rate in the period 30-500 ms after the event in question minus the average firing rate 0-1000 ms before the baseline event. In Experiment 1, the baseline event was uniformly defined as cue onset for each trial. In Experiment 2, baseline was calculated based on firing rate before lever extension for lever extension and auditory cue, and before light cue for light cue and reward, In Experiment 2, lever extensions were identical for all three trial types, so responses were aggregated across trial types to construct PSTHs and compute mean responses. For all events following lever press, responses were segregated by trial type (A vs. B vs. C).

The hazard rate was calculated as the probability of a lever press in a given bin given that no lever press has occurred – i.e. P($LP = t \mid LP >= t$)[35]. For each session, time following lever extension was binned in 100 ms intervals, and hazard rate in each bin was calculated as the percent of trials in which lever press occurred in that bin out of all trials in which lever press had not already occurred.

## Histology

After the final recordings, 100 mA current was passed through the recording electrodes to mark their locations. Rats were immediately sacrificed and perfused with cold phosphate-buffered saline (PBS) and 4% paraformaldehyde (PFA). Brains were dissected and stored for 24 h in PFA, followed by PBS. 100 um sections were collected and scanned on a slide-scanning microscope (Olympus) in 4x brightfield. Scans were compared with Paxinos & Watson's rat brain atlas to determine electrode placement (Supplementary Fig. 10)[75].

## Data analysis and statistics

All data analysis was carried out in MATLAB 2024a and GraphPad Prism 10. Residuals were tested for normality, and comparable non-parametric tests were substituted for their canonical parametric counterparts where appropriate. Outliers were detected and removed with Prism's ROUT method at Q = 1%. The significance level was set at 0.05. * indicates $p < 0.05$, **$p < 0.01$, ***$p < 0.001$, ****$p < 0.0001$.

Specific tests used were as follows: to compare a sample mean to a known reference value: one-sample $t$-test or one-sample Wilcoxon rank test. To compare sample means of two groups with paired observations: two-sample paired $t$-test or Wilcoxon signed-rank test. To compare sample means of two groups without paired observations: two sample unpaired $t$-test or Mann-Whitney $U$-test. To compare more than two groups with matched observations: 1-way repeated measures (RM) ANOVA or Friedman test. To compare more than two groups without matched observations: ordinary 1-way ANOVA or Kruskall-Wallis test. To compare more than two groups with matched observations along two different variables, 2-way RM ANOVA with Fisher's least significance difference (LSD) test for multiple comparisons, or 2-way RM mixed-effects analysis with Fisher's LSD test for multiple comparisons. To compare actual to expected categorical counts, Chi-squared test or Fisher's exact test.

## Reporting summary

Further information on research design is available in the Nature Portfolio Reporting Summary linked to this article.

# Data availability

All data for this study are available through Code Ocean at https://doi.org/10.24433/CO.0026587.v1. Source data are provided with this paper.

# Code availability

All code for this study is available through Code Ocean at https://doi.org/10.24433/CO.0026587.v1.

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

## Acknowledgements

We thank Y. Takahashi for support in building the electrode bundles and microdrives. We thank S. Ahmed, J. Berke, N. Eshel, H. Fields, P. Kalivas, S. Mahler, E. Margolis, V. Namboodiri, and Y. Shaham for comments and suggestions on the general conceptual framework and/or the manuscript. National Institutes of Health grant R00DA048085 (KM); National Institutes of Health intramural funding Z1ADA000587 (GS); German Research Foundation grant MA 8509/1-1 (KMC).

## Author contributions

Conceptualization: C.M.L., G.S., and K.M. Methodology: C.M.L., N.E.M., V.S.N., K.M.C., G.S., and K.M. Investigation: C.M.L., N.E.M., V.S.N., and K.M. Visualization: C.M.L., N.E.M., and K.M. Funding acquisition: G.S. and K.M. Project administration: K.M. Supervision: K.M. Writing – original draft: C.M.L. and K.M. Writing – review & editing: C.M.L., N.E.M., V.S.N., K.M.C., G.S., and K.M.

## Competing interests

The authors declare no competing interests.
