## [Transparent Peer Review file · Nature Communications]

Generalized cue reactivity in rat dopamine neurons after opioids

Corresponding Author: Dr Khaled Moussawi

Version 0:

Reviewer comments:

Reviewer #1

(Remarks to the Author)

In this manuscript, Lehmann and colleagues investigated how dopamine neurons respond to cues associated with natural rewards versus opioids in rats. Using behavioral tasks (both pavlovian and operant) and in vivo electrophysiology, they found that dopamine neuron firing was similarly enhanced in response to both drug-related and natural reward cues. Based on these findings, the authors suggested that while dopamine release is critical to cue reactivity, its role in response to drug cues may differ from previous theories. This manuscript is highly relevant, especially given its remarkable combination of complex behavioral analyses with advanced neuroscience techniques. The integration of these sophisticated methods offers valuable insights and demonstrates the strength of combining behavioral and neurophysiological approaches to explore the mechanisms underlying reward processing.

Comments to the authors:

- While the main results suggest that dopamine is involved in cue reactivity for drug versus non-drug rewards, the causality of this assumption was not directly tested. Recording from the VTA does not necessarily confirm that the signals originate from dopamine neurons or that dopamine release is involved in the specific behavioral tasks, as repeatedly implied by the authors. To address these concerns, the authors should perform direct causal manipulations to provide stronger evidence that their findings truly challenge the prevailing theories in the field. Additionally, if the authors aim to pinpoint the contribution of dopamine release, they should consider using other techniques designed for that purpose. Alternatively, I recommend that the authors focus on reporting and interpreting the data generated by the in vivo recording experiments.

- Comparing the neural results between the Pavlovian and operant tasks is challenging, further questioning the involvement of dopamine neurons in selective cue reactivity, especially given the absence of causal experiments. This challenge arises because the alternative non-drug rewards differ between the two experiments (sucrose in one and water in the other). The authors should address this concern and discuss the implications of using different rewards at both the behavioral and neural levels, particularly if they intend to make broader claims.

- Please consider including a supplemental notes within the main text to help readers follow the arguments regarding current theories of cue reactivity and the role of dopamine.

- Please consider avoiding the use of acronyms and abbreviations, as they can make the manuscript more difficult to read.

- Please provide a rationale for using only two female rats in Experiment 1 and only male rats in Experiment 2. Ideally, the experiments should include a balanced number of male and female rats to ensure that the findings are not biased by sex differences.

(Remarks on code availability)

Reviewer #2

(Remarks to the Author)

Lehmann and colleagues recorded from putative dopamine neurons in rats that were either opioid-naïve or opioid-exposed. They show that putative dopamine neurons in opioid-exposed rats had higher baseline firing rates and responded more vigorously to reward cues than putative dopamine neurons in opioid-naïve rats. Looking within the opioid-exposed group, there was no significant difference in putative dopamine responses to drug vs natural reward cues, even though the animals behaved differently in response to these cues. The authors interpret their data as evidence against “the prevailing hypothesis” of dopamine function in addiction.

My major concern with this paper is in the framing. The authors claim that the model of persistently positive (“non-compensable”) reward prediction error in response to drugs of abuse is “the leading” or “the prevailing” hypothesis in the field; thus, they argue that challenging this model is an important contribution. The problem is that I do not think this model is as accepted as the authors imply. As they state themselves in the introduction and in Supplementary Notes 1 and 7, multiple lines of evidence over >15 years have already disproved the model. By itself, then, pointing out further flaws in the model seems more of an incremental contribution to the literature, rather than a major advance.

The clearest result in the manuscript is that compared to opioid-naïve animals, opioid-exposed animals have higher dopamine firing rates (at baseline and in response to cues). I think the authors convincingly show this, but I believe this finding is largely known (for reviews, see Juarez & Han, 2016 or Mazei-Robison & Nestler, 2012). Demonstrating a direct relationship between this increase in firing rate (either baseline or cue-evoked) with a particular behavioral measure would be interesting, but would likely require within-animal experiments in which dopamine neuron activity is measured before and after opioid exposure. As it stands, it seems to me that the authors’ results actually argue against a strong role for cue-evoked dopamine in behavioral cue reactivity: their experiments revealed either no difference in dopamine neuron firing rates (Fig. 2) despite a difference in behavioral responses (Fig. 1h); or a difference in dopamine neuron firing rates (Fig. 5g) despite no difference in behavioral responses (Fig. 4h). The authors should comment on this dissociation.

The authors focus most on a putative null effect: no difference in dopamine responses between natural and drug reward cues. But my confidence in this finding is reduced by the fact that in the operant task, the authors are actually comparing dopamine responses to a cue predicting natural reward vs dopamine responses to a cue predicting natural plus drug rewards. As the authors state, the value of the latter is always more than the value of the former. Thus, finding no difference is contrary to many prior studies showing that reward-predicting cues elicit dopamine release in proportion to the magnitude of the predicted reward. The authors should comment on this discrepancy.

Minor concerns:

Pg. 3, line 16: Typo in “relapse”

Fig. 2g: is there a typo in the statistics that are included with the graph? The numbers are identical to Fig. 2f but the graph appears quite different, perhaps even showing higher sucrose responses than RMF responses

Please include rat sample size throughout, not just units or sessions. E.g., pg. 6, line 12, how many rats do these 77 units come from?

People (and rats) respond to opioid exposure in different ways. Not all will become ‘addicted’ in the classic sense. Could it be that the ‘null’ effect here (i.e., the lack of a difference in dopamine responses for cues predicting natural vs opioid reward) is because some or all of these rats never developed addictive-like behaviors? The authors should limit their conclusions to the effect of opioid exposure, rather than make claims about addiction, e.g., in pg. 4, line 5

Did the authors examine the putative GABA or other non-dopamine cell types that they recorded? Perhaps such an analysis could provide more novelty to the manuscript.

Is Extended Data Fig 3 accurate? The text refers to longer-timescale responses, but the graphed response appears short-lived.

(Remarks on code availability)

Reviewer #3

(Remarks to the Author)

This manuscript by Lehmann et al. aims to test of the hypothesis that opioids (and other abused drugs) dysregulate reinforcement learning due to the powerful pharmacological effects, which continue to trigger persistent dopamine neuron firing even when well predicted, leading to an over-attribution of value to drug-predictive cues. The researchers performed single unit recordings of VTA dopamine neuron firing in rats performing either Pavlovian (Exp 1) or discriminative instrumental (Exp 2) tasks, which in both cases included a within-subject comparison of auditory cues signaling either natural reward or intravenous remifentanyl reward. Both experiments found that VTA dopamine neuron firing was generally elevated during baseline periods and in response to natural rewards and auditory cues, relative to opioid-naïve control rats. Perhaps more importantly, this elevated firing was not specific to drug-predictive cues – similar elevations were apparent for natural reward and neutral cues. Indeed, dopamine neuron firing rates were not strongly modulated by cue-reward contingency and appeared to be divorced from cue-specific behavioral responses. The study addresses an important topic likely to be of broad interest to the neuroscience and addiction researchers. The main findings are exciting and somewhat surprising, raising basic questions about psychological significance of VTA dopamine neuron activity in addiction and beyond. The authors provide a thoughtful discussion of how their findings relate to the literature and leading theories of addiction. My critiques of the manuscript are itemized below.

1) It is not clear if there was any attempt to control for (counterbalance) perceptual differences across the auditory cues. The methods seem to indicate that these cues were confounded with different outcomes. In Experiment 2 this seems to have caused a difference in dopamine neuron firing rate that did not vary across drug-exposure groups. Nevertheless, confounding cues with contingencies makes it impossible to fully distinguish how these factors are influencing dopamine neuron firing, and tends to undermine within-subject cue comparisons, which is a major element of the study.

2) The main goal of this study was to test the prediction that drug-predictive cues acquire a uniquely strong capacity to elicit VTA dopamine neuron firing, which is thought to mediate their heightened behavioral reactivity relative to other reward-paired cues. There was however relatively modest analysis of the relationship between firing rates and cue reactivity. At a group level in Experiment 2, remifentanyl-cue specific behavioral responses are described, which is supposed to contrast with the lack of clear cue differences in firing rate. But this is somewhat problematic because there the drug-paired cue did elicit higher firing rates, even though this may very well be due to perceptual differences across auditory stimuli. It is difficult to gauge how these behavioral and electrophysiological effects align in size, and there is not much effort to show whether they correlate with each other across individuals/trials. The authors do show that one rat (#8) differed from others in the remifentanyl group, showing behavior and electrophysiological effects consistent with the control group. This is sensible but more should be done to evaluate the connection between behavior and dopamine firing rate.

3) Relatedly, it would be interesting to know if nonspecific elevated firing rate in opiate exposed rats was correlated with behavior. The principal component analysis in Fig 5L is suggestive but provides little insight into the nature of the relationship. The hypothesis under study would predict an association between Cue-B specific measures. In contrast, it would be equally interesting if baseline firing rates were associated with a general measure of motivation, such as the hazard rate for lever pressing.

4) Methods: the methods indicate that rats were water restricted in Experiment 1 even though the natural reward was 10% sucrose solution. Is this accurate? I know that restriction is need for the water reward in Experiment 2. I also found the description of the behavioral methods confusing for Experiment 2, particularly with regard to what 'correct responses' were and what the specific contingencies were. The schematic helped clarify but the text could use some attention, particularly page 36, lines 6-9.

5) The X2 and p values for Figures 2F and 2G are the same.

(Remarks on code availability)

Version 1:

Reviewer comments:

Reviewer #1

(Remarks to the Author)

The authors have addressed several of my initial concerns and acknowledged some limitations in their manuscript. I have no further comments. This will be an important manuscript for the field.

(Remarks on code availability)

Reviewer #2

(Remarks to the Author)

The authors did a commendable job expanding the discussion section to put their findings in context and spell out the limitations of their approach. I am more convinced now that their null finding between drug and natural reward cues will be broadly interesting to an audience of addiction neuroscientists. That said, I still find these results to be somewhat preliminary, because there is no clear and direct link between the neural recordings and behavior (as would be possible if they performed within-animal comparisons pre- vs post-opioid exposure, or engaged in causal experiments that manipulated dopamine neuron activity and examined the effects on drug cue reactivity). However, I understand that the authors are logistically limited from performing any additional experiments, and I am sympathetic to this. I only have two remaining suggestions:

1) if I understand the analysis in Fig 5L, this correlation could be driven by increased firing rates across-the-board and increased behavioral reactivity across-the-board. Furthermore, the positive result seems essentially guaranteed by the group differences that the authors already reported (i.e., the opioid-exposed group was already shown to have increases in both general behavioral reactivity and general dopamine neuron firing rates). It would be much more convincing if the correlation still exists when the analysis is restricted to JUST the opioid-exposed group. Is the correlation still there? Regardless, their correlation does not link recorded dopamine responses to drug-cue-specific reactivity per se. I would therefore temper the discussion of this correlation, both in the manuscript text and the abstract (second to last sentence). The findings do not convincingly show a direct link between generalized DA increases and specific drug-cue reactivity, so the authors should not claim that it does.

2) The authors should include a discussion in the manuscript about how the general increase in DA responses after opioid exposure is consistent with a large literature examining the effect of acute opioids (as they nicely summarized in their response to reviewers). The novelty here is the setting of chronic opioid exposure

(Remarks on code availability)

Reviewer #3

(Remarks to the Author)

The authors have adequately addressed my original concerns and I have not further feedback.

(Remarks on code availability)

We thank the reviewers for the thoughtful and articulate reviews of our manuscript. We have outlined below *specific responses to the raised concerns and suggestions*. The corresponding changes in the manuscript text are **highlighted in yellow**.

REVIEWER COMMENTS

Reviewer #1

- “In this manuscript, Lehmann and colleagues investigated how dopamine neurons respond to cues associated with natural rewards versus opioids in rats. Using behavioral tasks (both pavlovian and operant) and in vivo electrophysiology, they found that dopamine neuron firing was similarly enhanced in response to both drug-related and natural reward cues. Based on these findings, the authors suggested that while dopamine release is critical to cue reactivity, its role in response to drug cues may differ from previous theories. This manuscript is highly relevant, especially given its remarkable combination of complex behavioral analyses with advanced neuroscience techniques. The integration of these sophisticated methods offers valuable insights and demonstrates the strength of combining behavioral and neurophysiological approaches to explore the mechanisms underlying reward processing.”

We thank the reviewer for their positive assessment of the significance and impact of this manuscript.

- “While the main results suggest that dopamine is involved in cue reactivity for drug versus non-drug rewards, the causality of this assumption was not directly tested. Recording from the VTA does not necessarily confirm that the signals originate from dopamine neurons or that dopamine release is involved in the specific behavioral tasks, as repeatedly implied by the authors. To address these concerns, the authors should perform direct causal manipulations to provide stronger evidence that their findings truly challenge the prevailing theories in the field. Additionally, if the authors aim to pinpoint the contribution of dopamine release, they should consider using other techniques designed for that purpose. Alternatively, I recommend that the authors focus on reporting and interpreting the data generated by the in vivo recording experiments.”

The reviewer is right to emphasize that we did not directly manipulate dopamine neuronal firing or measure dopamine release (though the activity-based screening is grounded in much prior work to reliably identify midbrain dopamine neurons based on spiking activity). We have revised our manuscript in the Abstract and pp. 18-19 to clarify that this was not part of our experimental design, and to focus more directly on the results at hand. However, we have good reason to believe that the population we recorded is indeed dopaminergic (Fig. 1, 2, Extended data Fig. 2). The evidence for dopamine’s importance in both Pavlovian and operant conditioning as well as addiction is substantial and is a shared assumption held by much of the field (see Berke & Hyman, 2000; Di Chiara & Bassareo, 2007; Solinas et al., 2019; Wise and Robble, 2020, Samaha et al. 2021, but see Nutt et al., 2015 for a contrasting view)¹⁻⁶. Thus, we believe that to completely exclude these considerations from our discussion of the implications of our findings would be a gross omission. Accordingly, in our revision we have attempted to

make it clear to readers that our data suggest a variety of avenues for future research, but do not demonstrate a proposed alternative role for dopamine.

In addition, since completing the experiments described in this manuscript, the senior author (Dr. Khaled Moussawi) has taken up a new position at UCSF, and we are not well-positioned to carry out additional experiments at this time. Thus, though here and elsewhere important points are raised which are worthy of further study, and we believe would be of sufficient importance to warrant independent investigation, we are not currently able to perform such experiments. Nevertheless, we hope to convey in our responses and in the manuscript text that the results we report are important enough to publish without this additional work, with the helpful limitations and qualifications suggested by the reviewers to help readers place them in the correct context.

- “Comparing the neural results between the Pavlovian and operant tasks is challenging, further questioning the involvement of dopamine neurons in selective cue reactivity, especially given the absence of causal experiments. This challenge arises because the alternative non-drug rewards differ between the two experiments (sucrose in one and water in the other). The authors should address this concern and discuss the implications of using different rewards at both the behavioral and neural levels, particularly if they intend to make broader claims.”

In Experiment 1, sucrose reward was used rather than water because it is generally more rewarding for rats (see e.g., Brennan et al., 2001)⁷. Because the tested hypothesis predicted that the drug cue response would be greater than the natural reward cue response regardless of value, we wanted a highly rewarding natural reward to minimize the possibility of observing the predicted pattern simply because of a lower relative value of the natural reward (clarified on p.5). In Experiment 2, we adopted a different approach. Because drug and natural reward value as inferred from dopamine neuron firing cannot be directly compared (discussed in manuscript p. 5, lines 9-13) the only way to definitively establish the relative value of the two conditions was to compare natural reward alone to natural reward plus drug, which, according to the initial cue reactivity hypothesis, should have greater value regardless of the specific reward values of the natural reward and drug separately. For this approach, the added reward value of sucrose over water was unnecessary, so we chose to use water which simplifies daily preparation and cleaning of equipment. As such, these experiments tested different facets of the cue reactivity hypothesis. Furthermore, we do not believe that the specific firing responses to cues and rewards administered to different animals under different experimental conditions at different institutions can be directly compared except in the broadest of terms. Thus, it was never our intention to directly compare the natural reward responses in Experiment 1 with those in Experiment 2. Overall, the spiking responses to both rewards are similar to what has been observed by other investigators under similar conditions (Sadacca et al., 2016)⁸, but the central claims of our paper do not depend on equivalence or any specific comparison between the natural rewards in experiments 1 and 2.

- “Please consider including the supplemental notes within the main text to help readers follow the arguments regarding current theories of cue reactivity and the role of dopamine.”

Where possible, we have incorporated our notes into the main text and highlighted them in yellow (pp. 3-4,6,11-17).

- “Please consider avoiding the use of acronyms and abbreviations, as they can make the manuscript more difficult to read.”

To improve readability, we have replaced acronyms and abbreviations where possible. To avoid cluttering the revised manuscript, we have logged these changes with tracked changes rather than highlighting in yellow.

- “Please provide a rationale for using only two female rats in Experiment 1 and only male rats in Experiment 2. Ideally, the experiments should include a balanced number of male and female rats to ensure that the findings are not biased by sex differences.”

Experiment 1 was initially planned to include a balanced number of male and female rats. However, significantly greater attrition was seen in female rats due to catheter occlusion, complications of cranial implantation surgery, and electrode failures. Female rats also generally perform fewer trials and are more affected by the weight of cranial implants. Thus, to avoid further complications, the experiment was completed with male rats. Experiment 2 was completed in the Schoenbaum lab, which generally uses male rats primarily due to the exact concerns described above. Our experiments were not intended to investigate sex differences; nevertheless, we recognize that the lack of sex balancing remains an important limitation. We have revised the discussion section in the manuscript on p. 19 to emphasize this limitation to the reader.

Reviewer #2

- “Lehmann and colleagues recorded from putative dopamine neurons in rats that were either opioid-naïve or opioid-exposed. They show that putative dopamine neurons in opioid-exposed rats had higher baseline firing rates and responded more vigorously to reward cues than putative dopamine neurons in opioid-naïve rats. Looking within the opioid-exposed group, there was no significant difference in putative dopamine responses to drug vs natural reward cues, even though the animals behaved differently in response to these cues. The authors interpret their data as evidence against “the prevailing hypothesis” of dopamine function in addiction. My major concern with this paper is in the framing. The authors claim that the model of persistently positive (“non-compensable”) reward prediction error in response to drugs of abuse is “the leading” or “the prevailing” hypothesis in the field; thus, they argue that challenging this model is an important contribution. The problem is that I do not think this model is as accepted as the authors imply. As they state themselves in the introduction and in Supplementary Notes 1 and 7, multiple lines of evidence over >15 years have already disproved the model. By itself, then, pointing out further flaws in the model seems more of an incremental contribution to the literature, rather than a major advance.”

We are sympathetic to the reviewer’s perspective on the existing behavioral evidence from blocking experiments^{9,10} (published in 2007 and 2010), which question some tenants of the PRE-based cue reactivity hypothesis. As the reviewer points out, we acknowledged that certain predictions of the “noncompensable” hypothesis have not been supported by experimental results (discussed in the introduction on p.3). However, we respectfully disagree with the reviewer’s estimation of the significance of the experiments in this manuscript or the continued impact of the noncompensable model we challenge here.

It is our perspective that incommensurate findings that do not directly test the central hypothesis of a model are not always sufficient to dislodge it, particularly when few plausible alternatives have been proposed. Most formal models of the pathophysiology of drug addiction do not address the cue reactivity phenomenon at all, and those that do, to our knowledge, are generally in agreement with the “noncompensable” hypothesis and its proposition of selective enhancement of the dopaminergic response to drug cues (e.g., Berridge and Robinson’s incentive salience model (2016)¹¹, Di Chiara’s associative learning disorder model (1999)¹², Leyton and Vezina’s “integrative” model (2014)¹³). Importantly, the noncompensable model is still presented as the default mechanism underlying cue reactivity in major recent reviews on the neurobiology of drug addiction by leaders in the reward and addiction fields (e.g., Keiflin and Janak (2015), p. 253¹⁴ (see below copied Figure from the review), Luscher and Janak (2021), p. 176¹⁵).

Finally, to our knowledge, no prior experiments directly compared dopamine neuronal responses to cues predicting drugs vs. natural rewards which is very important for the addiction research field, independent of the model it seeks to challenge. Together, this knowledge gap, the blocking experiments’ evidence against the noncompensable model, and the continued interest in the field in this model, all motivated our work in this study which revealed several novel and surprising findings.

[REDACTED]

Figure copied from Keiflin and Janak (2015) highlighting the persisting impact of the noncompensable reward prediction error hypothesis of cue reactivity in the drug addiction field.

- “The clearest result in the manuscript is that compared to opioid-naïve animals, opioid-exposed animals have higher dopamine firing rates (at baseline and in response to cues). I think the authors convincingly show this, but I believe this finding is largely known (for reviews, see Juarez & Han, 2016 or Mazei-Robison & Nestler, 2012).”

- *We appreciate the reviewer’s confidence in our findings. However, we disagree with their perspective on its novelty as we don’t think that increased baseline firing rate in dopamine neurons, in vivo, in the absence of acute opioids has been shown previously. The highlighted reviews in the comment above refer to results from experiments that differ in crucial ways from those presented in this manuscript. The vast majority of the effects described in these reviews are due to acute pharmacological effects of opioids (acute opioids pharmacologically increase firing of dopamine neurons through disinhibition). Two primary research papers could appear to fall outside this category, as they purport to examine the effects of chronic morphine rather than acute morphine administration (Mazei-Robison et al., (2011)¹⁶; Georges et al., (2006)¹⁷). However, in both cases, the results showing enhanced dopamine activity were obtained from animals implanted with morphine pellets that slowly release a constant supply of morphine over weeks and the recordings were not obtained in the clear absence of acute morphine effects. Thus, the electrophysiological findings could be attributed to the presence of morphine onboard (in vivo or in slice, which would still contain morphine released from the pellet prior to harvesting of the brains). In contrast, our data show elevated firing rates before drug is administered (Experiment 1) or during sessions without any drug at all (Experiment 2), showing increased tonic firing that persists at least temporarily without drug on board. Moreover, cue-responsive phasic firing, which is physiologically distinct from tonic firing, critical to reward learning, and hypothesized to be critical to cue reactivity was not investigated by any of the referenced papers (the bulk of our manuscript is concerned with this particular set of findings).*

- “Demonstrating a direct relationship between this increase in firing rate (either baseline or cue-evoked) with a particular behavioral measure would be interesting, but would likely require

within-animal experiments in which dopamine neuron activity is measured before and after opioid exposure.”

We appreciate this important point raised by the reviewer. It is hard to firmly conclude a causal relationship between firing and drug exposure with our current design. As the reviewer points out, it would be more impactful to measure the firing of dopamine neurons before and after opioid exposure. However, such design is exceedingly difficult in our model (single unit recording from dopamine neurons in the rat) as the dopamine neurons yield of our recording sessions is unpredictable, generally very low (at most 1-2 neurons per sessions), and cannot be determined in real-time (neuronal identification is done through automated clustering at the end of the experiment). As such, it is not feasible to compare dopamine firing within-animal pre- vs. post-opioid exposure. A more plausible model for such comparison would be using fiber photometry to measure dopamine release. As we won't be able to conduct such experiments at this point, we have added language in the main text (p. 17) to emphasize the limitation of our design regarding the interpretation of the difference in tonic firing data.

- “As it stands, it seems to me that the authors’ results actually argue against a strong role for cue-evoked dopamine in behavioral cue reactivity: their experiments revealed either no difference in dopamine neuron firing rates (Fig. 2) despite a difference in behavioral responses (Fig. 1h); or a difference in dopamine neuron firing rates (Fig. 5g) despite no difference in behavioral responses (Fig. 4h). The authors should comment on this dissociation.”

We agree that our findings suggest that the relative dopamine firing responses to drug vs. non-drug cues does not explain differences in subsequent motivated response to these cues, as the noncompensable reward hypothesis would suggest. Note that in experiment 1, there was no comparable “retrieval” behavior to obtain drug reward that could plausibly be compared to retrieval from the sucrose port, so such comparisons cannot be rigorously tested in this dataset; however, in the supplemental progressive ratio task in Experiment 2 (Extended Fig. 4), the behavior was consistent with greater motivation to obtain remifentanyl than sucrose at the doses used in Experiment 1. In Experiment 2, both remifentanyl-specific (Fig. 4f-j) and non-specific (Fig. 4k) measures of behavior suggested increased motivation in the remifentanyl group, and to remifentanyl-predicting cues compared to controls. Simultaneously, we found increased dopamine firing across all cues in remifentanyl-exposed compared to drug-naïve rats, and these findings were correlated (Fig. 5l), including the finding that the lone remifentanyl-exposed rat that did not follow this behavioral pattern also lacked elevated dopamine responses. Thus, while we do think these data suggest a dissociation between cue response and motivational value (see our further response below), there remains a broader connection between dopamine activity across cues and task states and behavior. As we have acknowledged elsewhere in our responses, we cannot establish causality, but the correlation nevertheless strikes us as worthy of consideration and further investigation. We now clarify this further in our discussion on pages 18 and 19.

- “The authors focus most on a putative null effect: no difference in dopamine responses between natural and drug reward cues. But my confidence in this finding is reduced by the fact that in the

operant task, the authors are actually comparing dopamine responses to a cue predicting natural reward vs dopamine responses to a cue predicting natural plus drug rewards. As the authors state, the value of the latter is always more than the value of the former. Thus, finding no difference is contrary to many prior studies showing that reward-predicting cues elicit dopamine release in proportion to the magnitude of the predicted reward. The authors should comment on this discrepancy.”

We agree that the disconnect between objective reward value and dopamine firing response is novel and surprising, and importantly, directly challenges the noncompensable reward prediction error hypothesis of cue reactivity, rendering our findings highly significant for the reinforcement learning and addiction fields. As the reviewer indicates, our findings contradict the standard temporal difference model which describes how in normal learning, predictive cues produce dopamine responses that reflect the magnitude of the change in expected value from predicting the associated reward. However, this framework was developed based on data obtained from animals behaving to receive natural rewards; drug rewards, which bypass systems for evaluating reward value to directly interact with the midbrain dopamine system may not result in the same outcomes. This is what we believe our data show. As we clarify in the discussion (p. 15-16), we believe there's a generalized reinforcement gain caused by opioids that increase dopaminergic response to any and all cues (discrete and contextual), even those that do not necessarily predict the drug, which has significant clinical implications related to the incentive salience to any cues contextually associated with drugs including non-drug predicting ones. Such a framework is consistent with the improved clinical outcomes and high abstinence rates in substance use disorders patients who are discharged into new environments, not associated with the previous drug context¹⁸ (added to the discussion on p. 20).

Minor concerns:

- Pg. 3, line 16: Typo in “relapse”

Typo corrected.

- “Fig. 2g: is there a typo in the statistics that are included with the graph? The numbers are identical to Fig. 2f but the graph appears quite different, perhaps even showing higher sucrose responses than RMF responses”

In both panels, the Chi-squared analysis compared the number of units that respond more to the remifentanyl cue than the sucrose cue during the interval of interest. Even though the firing responses are different between the 2 panels, the number of neurons that responded more to remifentanyl than sucrose cues during interval one was the same as the number responding more to remifentanyl than sucrose cues during interval two; therefore, the same Chi-squared statistic was computed. We now clarify this in the legend.

- “Please include rat sample size throughout, not just units or sessions. E.g., pg. 6, line 12, how many rats do these 77 units come from?”

Detailed N is now included on pages 7,12.

- “People (and rats) respond to opioid exposure in different ways. Not all will become ‘addicted’ in the classic sense. Could it be that the ‘null’ effect here (i.e., the lack of a difference in dopamine responses for cues predicting natural vs opioid reward) is because some or all of these rats never developed addictive-like behaviors? The authors should limit their conclusions to the effect of opioid exposure, rather than make claims about addiction, e.g., in pg. 4, line 5.”

We agree with the reviewer that drug exposure is very different than addiction. In Experiment 2, we observed the null effect within-cell for all rats, but showed behavioral changes consistent with an addiction model, suggesting that the null effect is not due to a lack of susceptibility to the drug. However, we acknowledge that this was not directly tested in the Pavlovian design of Experiment 1 and have edited our language in the main text on page 19 to emphasize the applicability of our data to opioid exposure over addiction.

- “Did the authors examine the putative GABA or other non-dopamine cell types that they recorded? Perhaps such an analysis could provide more novelty to the manuscript.”

The data on non-dopamine cells are being analyzed and will be reported in a different manuscript.

- “Is Extended Data Fig 3 accurate? The text refers to longer-timescale responses, but the graphed response appears short-lived.”

The short response in this figure is the phasic cue response. The extended response in this figure is obtained after months of training, thus it is to be expected that the rats have achieved quite a high tolerance and consequently, a direct drug effect only slightly above baseline. For earlier drug response, see Fig. 1. This is now clarified in the figure legend.

Reviewer #3

“This manuscript by Lehmann et al. aims to test of the hypothesis that opioids (and other abused drugs) dysregulate reinforcement learning due to the powerful pharmacological effects, which continue to trigger persistent dopamine neuron firing even when well predicted, leading to an over-attribution of value to drug-predictive cues. The researchers performed single unit recordings of VTA dopamine neuron firing in rats performing either Pavlovian (Exp 1) or discriminative instrumental (Exp 2) tasks, which in both cases included a within-subject comparison of auditory cues signaling either natural reward or intravenous remifentanil reward. Both experiments found that VTA dopamine neuron firing was generally elevated during baseline periods and in response to natural rewards and auditory cues, relative to opioid-naïve control rats. Perhaps more importantly, this elevated firing was not specific to drug-predictive cues – similar elevations were apparent for natural reward and neutral cues. Indeed, dopamine neuron firing rates were not strongly modulated by cue-reward contingency and appeared to be divorced from cue-specific behavioral responses. The study addresses an important topic likely to be of broad interest to the neuroscience and addiction researchers. The main findings are exciting and somewhat surprising, raising basic questions about psychological significance of VTA dopamine neuron activity in addiction and beyond. The authors provide a thoughtful discussion of how their findings relate to the literature and leading theories of addiction. My critiques of the manuscript are itemized below.”

We thank the reviewer for their positive assessment of the significance and impact of this manuscript.

- “It is not clear if there was any attempt to control for (counterbalance) perceptual differences across the auditory cues. The methods seem to indicate that these cues were confounded with different outcomes. In Experiment 2 this seems to have caused a difference in dopamine neuron firing rate that did not vary across drug-exposure groups. Nevertheless, confounding cues with contingencies makes it impossible to fully distinguish how these factors are influencing dopamine neuron firing, and tends to undermine within-subject cue comparisons, which is a major element of the study.”

- *The reviewer is correct to note that cues were not counterbalanced to control for perceptual differences due to the sensory properties of these cues. This is indeed a significant limitation, and we have added language to the discussion on page 19 to highlight this limitation for readers. Nevertheless, we believe that the consistency of our results across the two behavioral tasks, different cue types within-experiment, two institutions, etc. are powerful evidence that our results are not artifacts of perceptual differences across cues.*

- “The main goal of this study was to test the prediction that drug-predictive cues acquire a uniquely strong capacity to elicit VTA dopamine neuron firing, which is thought to mediate their heightened behavioral reactivity relative to other reward-paired cues. There was however relatively modest analysis of the relationship between firing rates and cue reactivity. At a group level in Experiment 2, remifentanil-cue specific behavioral responses are described, which is supposed to contrast with the lack of clear cue differences in firing rate. But this is somewhat problematic because there the drug-paired cue did elicit higher firing rates, even though this may

very well be due to perceptual differences across auditory stimuli. It is difficult to gauge how these behavioral and electrophysiological effects align in size, and there is not much effort to show whether they correlate with each other across individuals/trials. The authors do show that one rat (#8) differed from others in the remifentanyl group, showing behavior and electrophysiological effects consistent with the control group. This is sensible but more should be done to evaluate the connection between behavior and dopamine firing rate. Relatedly, it would be interesting to know if nonspecific elevated firing rate in opiate exposed rats was correlated with behavior. The principal component analysis in Fig 5L is suggestive but provides little insight into the nature of the relationship. The hypothesis under study would predict an association between Cue-B specific measures. In contrast, it would be equally interesting if baseline firing rates were associated with a general measure of motivation, such as the hazard rate for lever pressing.”

The reviewer raises an important point regarding the relationship between neuronal firing and behavior. We did not see a significant relationship between the average baseline firing per session and the hazard rate. We also investigated the relation between baseline firing and the number and probability of correct responses and congruent entries following cues A and B and similarly found no significant relationship. Illustrative panels are included below. This is likely due to the inherent limitation of single-unit recordings from dopamine neurons. It is difficult to connect neural activity to behavior when our yield is very low during each recording session. Since any effect on behavior will be dependent on population activity, any given neuron (and we typically are only able to record 1-2 putative dopamine neurons, if any, in a given session) is a noisy measure of the population activity, making granular connections between behavior and neural activity difficult to substantiate without extremely large datasets. Given the limitations of single-unit electrophysiology, we now acknowledge that these specific behavioral effects remain important open questions to be answered by future research and have added language on pages 17 and 19 to emphasize this to the reader.

With respect to phasic activation, with upwards of ten measurements of different behaviors and intervals of neural activity, we opted to avoid large numbers of potentially spurious correlations, and instead attempted to distill our results to the essential finding through the PCs in Fig. 5l. In our behavior PCA, log ratio of congruent feeder entries and hazard rate loaded strongly on PC1 (see table below), which was unsurprising, as these measures showed a starker difference between drug exposed vs. naïve rats. Similarly, putative dopamine firing to lever extension, light cue, and reward loaded more strongly onto PC1 than did response to auditory cues A and B, most likely due to the increased variability due to enhanced firing in the opioid-exposed group (Fig. 4g-i). Generally, measures that had large variability (i.e., loaded strongly on respective PC1's) tended to correlate strongly and positively, and those with more restricted variability correlated weakly or insignificantly. Thus, to convey approximately the same information with one panel and statistical test, we elected to report the PC correlation instead.

BEHAVIOR	PC1 loadings	NEURONS	PC1 loadings
log (congruent B entries/congruent A entries)	0.7150	Lever extension firing (Hz-baseline)	0.4984
log (% B correct/ % A correct)	0.2808	Cue A/B firing (Hz-baseline)	0.1538
P (congruent 1st entry after cue B) / P (congruent 1st entry after cue A)	0.2204	Light A/B firing (Hz-baseline)	0.7082
Hazard rate	0.6012	Reward A/B firing (Hz-baseline)	0.4758

- “Methods: the methods indicate that rats were water restricted in Experiment 1 even though the natural reward was 10% sucrose solution. Is this accurate? I know that restriction is need for the water reward in Experiment 2. I also found the description of the behavioral methods confusing for Experiment 2, particularly with regard to what ‘correct responses’ were and what the specific contingencies were. The schematic helped clarify but the text could use some attention, particularly page 36, lines 6-9.”

Rats were water-restricted in both experiments to maintain high levels of task engagement to complete large number of trials. We have revised our description of the task on pages 5 and 9 to improve clarity of the methods description of Experiment 2.

- “The X2 and p values for Figures 2F and 2G are the same.”

In both panels, the Chi-squared analysis compared the number of units that respond more to the remifentanil cue than the sucrose cue during the interval of interest. Even though the firing responses are different between the 2 panels, the number of neurons that responded more to remifentanil than sucrose cues during interval one was the same as the number responding more to remifentanil than sucrose cues during interval two; therefore, the same Chi-squared statistic was computed. This is now clarified in the legend.

REFERENCES

- 1 Berke, J. D. & Hyman, S. E. Addiction, dopamine, and the molecular mechanisms of memory. *Neuron* **25**, 515-532 (2000). [https://doi.org/10.1016/s0896-6273\(00\)81056-9](https://doi.org/10.1016/s0896-6273(00)81056-9)
- 2 Di Chiara, G. & Bassareo, V. Reward system and addiction: what dopamine does and doesn't do. *Curr Opin Pharmacol* **7**, 69-76 (2007).
<https://doi.org/10.1016/j.coph.2006.11.003>
- 3 Wise, R. A. & Robble, M. A. Dopamine and Addiction. *Annu Rev Psychol* **71**, 79-106 (2020). <https://doi.org/10.1146/annurev-psych-010418-103337>
- 4 Solinas, M., Belujon, P., Fernagut, P. O., Jaber, M. & Thiriet, N. Dopamine and addiction: what have we learned from 40 years of research. *J Neural Transm* **126**, 481-516 (2019).
<https://doi.org/10.1007/s00702-018-1957-2>
- 5 Samaha, A. N., Khoo, S. Y., Ferrario, C. R. & Robinson, T. E. Dopamine 'ups and downs' in addiction revisited. *Trends Neurosci* **44**, 516-526 (2021).
<https://doi.org/10.1016/j.tins.2021.03.003>
- 6 Nutt, D. J., Lingford-Hughes, A., Erritzoe, D. & Stokes, P. R. The dopamine theory of addiction: 40 years of highs and lows. *Nat Rev Neurosci* **16**, 305-312 (2015).
<https://doi.org/10.1038/nrn3939>
- 7 Brennan, K., Roberts, D. C., Anisman, H. & Merali, Z. Individual differences in sucrose consumption in the rat: motivational and neurochemical correlates of hedonia. *Psychopharmacology (Berl)* **157**, 269-276 (2001).
<https://doi.org/10.1007/s002130100805>
- 8 Sadacca, B. F., Jones, J. L. & Schoenbaum, G. Midbrain dopamine neurons compute inferred and cached value prediction errors in a common framework. *Elife* **5** (2016).
<https://doi.org/10.7554/eLife.13665>
- 9 Panlilio, L. V., Thorndike, E. B. & Schindler, C. W. Blocking of conditioning to a cocaine-paired stimulus: testing the hypothesis that cocaine perpetually produces a signal of larger-than-expected reward. *Pharmacol Biochem Behav* **86**, 774-777 (2007).
<https://doi.org/10.1016/j.pbb.2007.03.005>
- 10 Marks, K. R., Kearns, D. N., Christensen, C. J., Silberberg, A. & Weiss, S. J. Learning that a cocaine reward is smaller than expected: A test of Redish's computational model of addiction. *Behav Brain Res* **212**, 204-207 (2010).
<https://doi.org/10.1016/j.bbr.2010.03.053>
- 11 Berridge, K. C. & Robinson, T. E. Liking, wanting, and the incentive-sensitization theory of addiction. *Am Psychol* **71**, 670-679 (2016). <https://doi.org/10.1037/amp0000059>
- 12 Di Chiara, G. Drug addiction as dopamine-dependent associative learning disorder. *Eur J Pharmacol* **375**, 13-30 (1999). [https://doi.org/10.1016/s0014-2999\(99\)00372-6](https://doi.org/10.1016/s0014-2999(99)00372-6)
- 13 Leyton, M. & Vezina, P. Dopamine ups and downs in vulnerability to addictions: a neurodevelopmental model. *Trends Pharmacol Sci* **35**, 268-276 (2014).
<https://doi.org/10.1016/j.tips.2014.04.002>
- 14 Keiflin, R. & Janak, P. H. Dopamine Prediction Errors in Reward Learning and Addiction: From Theory to Neural Circuitry. *Neuron* **88**, 247-263 (2015).
<https://doi.org/10.1016/j.neuron.2015.08.037>
- 15 Luscher, C. & Janak, P. H. Consolidating the Circuit Model for Addiction. *Annu Rev Neurosci* **44**, 173-195 (2021). <https://doi.org/10.1146/annurev-neuro-092920-123905>

- 16 Mazei-Robison, M. S. *et al.* Role for mTOR signaling and neuronal activity in morphine-induced adaptations in ventral tegmental area dopamine neurons. *Neuron* **72**, 977-990 (2011). <https://doi.org/10.1016/j.neuron.2011.10.012>
- 17 Georges, F., Le Moine, C. & Aston-Jones, G. No effect of morphine on ventral tegmental dopamine neurons during withdrawal. *J Neurosci* **26**, 5720-5726 (2006). <https://doi.org/10.1523/JNEUROSCI.5032-05.2006>
- 18 Jason, L. A., Olson, B. D., Ferrari, J. R. & Lo Sasso, A. T. Communal housing settings enhance substance abuse recovery. *Am J Public Health* **96**, 1727-1729 (2006). <https://doi.org/10.2105/AJPH.2005.070839>

We thank the Editor and reviewers for their positive feedback on our revised manuscript. Please find below our response to the remaining concerns.

REVIEWER COMMENTS

Reviewer #1

“The authors have addressed several of my initial concerns and acknowledged some limitations in their manuscript. I have no further comments. This will be an important manuscript for the field.”

We thank the reviewer for their positive feedback.

Reviewer #2

“The authors did a commendable job expanding the discussion section to put their findings in context and spell out the limitations of their approach. I am more convinced now that their null finding between drug and natural reward cues will be broadly interesting to an audience of addiction neuroscientists. That said, I still find these results to be somewhat preliminary, because there is no clear and direct link between the neural recordings and behavior (as would be possible if they performed within-animal comparisons pre- vs post-opioid exposure, or engaged in causal experiments that manipulated dopamine neuron activity and examined the effects on drug cue reactivity). However, I understand that the authors are logistically limited from performing any additional experiments, and I am sympathetic to this.

We thank the reviewer for their positive feedback.

“If I understand the analysis in Fig 5L, this correlation could be driven by increased firing rates across-the-board and increased behavioral reactivity across-the-board. Furthermore, the positive result seems essentially guaranteed by the group differences that the authors already reported (i.e., the opioid-exposed group was already shown to have increases in both general behavioral reactivity and general dopamine neuron firing rates). It would be much more convincing if the correlation still exists when the analysis is restricted to JUST the opioid-exposed group. Is the correlation still there? Regardless, their correlation does not link recorded dopamine responses to drug-cue-specific reactivity per se. I would therefore temper the discussion of this correlation, both in the manuscript text and the abstract (second to last sentence). The findings do not convincingly show a direct link between generalized DA increases and specific drug-cue reactivity, so the authors should not claim that it does.”

Per the reviewer's suggestion, we performed the same correlation analysis between the principal components of behavior and dopamine firing in only opioid-exposed rats. This showed a positive correlation (Spearman's $\rho = 0.367$, $n = 22$, $t(20) = 1.768$) that approached our significance threshold ($p = 0.059$). This was added to the Results section (pg. 13). We also modified the abstract text as suggested.

“The authors should include a discussion in the manuscript about how the general increase in DA responses after opioid exposure is consistent with a large literature examining the effect of acute opioids (as they nicely summarized in their response to reviewers). The novelty here is the setting of chronic opioid exposure”.

Per the reviewer’s suggestion, we have now added a discussion (with the relevant references) on the prior literature on acute opioid enhancement of dopamine neuronal firing (pg. 16).

Reviewer #3

“The authors have adequately addressed my original concerns and I have no further feedback.”

We thank the reviewer for their positive feedback.